# An Extended Usability and UX Evaluation of a Mobile Application for the Navigation of Individuals with Blindness and Visual Impairments Outdoors—An Evaluation Framework Based on Training

**DOI:** 10.3390/s22124538

**Published:** 2022-06-16

**Authors:** Paraskevi Theodorou, Kleomenis Tsiligkos, Apostolos Meliones, Costas Filios

**Affiliations:** Department of Digital Systems, School of Information and Communication Technologies, University of Piraeus, 18534 Piraeus, Greece; ktsiligkos@unipi.gr (K.T.); meliones@unipi.gr (A.M.); costasfilios@unipi.gr (C.F.)

**Keywords:** UX (user experience), usability, mobile app, user-centered training, visual impairments and blindness

## Abstract

Navigation assistive technologies have been designed to support the mobility of people who are blind and visually impaired during independent navigation by providing sensory augmentation, spatial information and general awareness of their environment. This paper focuses on the extended Usability and User Experience (UX) evaluation of BlindRouteVision, an outdoor navigation smartphone application that tries to efficiently solve problems related to the pedestrian navigation of visually impaired people without the aid of guides. The proposed system consists of an Android application that interacts with an external high-accuracy GPS sensor tracking pedestrian mobility in real-time, a second external device specifically designed to be mounted on traffic lights for identifying traffic light status and an ultrasonic sensor for detecting near-field obstacles along the route of the blind. Moreover, during outdoor navigation, it can optionally incorporate the use of Public Means of Transport, as well as provide multiple other uses such as dialing a call and notifying the current location in case of an emergency. We present findings from a Usability and UX standpoint of our proposed system conducted in the context of a pilot study, with 30 people having varying degrees of blindness. We also received feedback for improving both the available functionality of our application and the process by which the blind users learn the features of the application. The method of the study involved using standardized questionnaires and semi-structured interviews. The evaluation took place after the participants were exposed to the system’s functionality via specialized user-centered training sessions organized around a training version of the application that involves route simulation. The results indicate an overall positive attitude from the users.

## 1. Introduction

Globally, vision loss is the third most frequent disability. The incidence of eyesight loss is expected to climb as life expectancy and population growth increase globally. Although the modeling projections have limitations, the extended increase in prevalence and the global numbers of blindness and vision impairment (such as the number of blind people increasing to 38.5 million by 2020 and 115 million by 2050) demonstrate the magnitude of the challenge [1]. For the next three decades, it is projected that individuals with moderate to severe visual impairment will increase to more than 550 million people, up from approximately 200 million individuals in 2020 [2]. According to the World Health Organization (WHO), about 90% of people living with Moderate and Severe Visual Impairments (MSVI) are in low-income regions, raising serious issues for social and economic growth in these areas [3].

The increase in visual impairment and blindness in both developing and developed nations can be attributed to several reasons, including inheritance, accidents, illnesses and the ageing population, while the severity varies by gender, the individual’s income and the country’s economic status. In practice, the latter largely influences the way of tackling such a crisis as it enables access to the most up-to-date vision correction research or medication that could have prevented or reduced cases of this disability in the first place. Although there are cost-effective therapies for preventing or treating most types of vision loss, their availability varies greatly between nations and regions. The relationship between vision loss and socioeconomic factors might help with public health planning. Furthermore, complementary to addressing these challenges is the use of various assistive technologies [4].

The inconveniences of the current assistive devices for the blind and visually impaired and the inter-related socio-psychological attributes [5] had always been an area of great interest for various researchers around the world as some of these technologies are too technical, not portable or impractical to use. Advancements in technology allow for the improvement of these limitations. Despite significant progress and a wide range of technical solutions that resulted in several frameworks, navigation assistance devices are still not extensively utilized, user approval is poor [6] and many of them, as described in some very interesting literature reviews covering blind navigation [7,8,9], are restricted in scope.

In this paper, we present a blind navigation system developed as part of the MANTO [10,11,12] project, called BlindRouteVision, that enables users to safely navigate to destinations with an option to include Public Means of Transport, its Usability and UX evaluation tailored to the special needs of this target group, and the specialized training tool and sessions developed to increase the acceptance of the proposed solution that conforms to the basic theories, principles and merits of Special Needs Education. BlindRouteVision also enables the passing of marked pedestrian crossings near traffic lights with zero latency notifications, as well as detecting obstacles in the user’s path with the aid of a custom-made ultrasonic-based obstacle detection system. In Section 1.1, we present past and current solutions to the problem of navigation for people with blindness and visual impairments. Section 1.2 presents the architecture and the basic principles of operation of our proposed solution along with the design process we employed to produce the artefact. Subsequently, Section 2.1, Section 2.2, Section 2.3, Section 2.4 and Section 2.5 present the methodology with which we conducted the usability and UX evaluation, while in Section 3.1, Section 3.2, Section 3.3, Section 3.4, Section 3.5, Section 3.6 and Section 3.7, we present the data findings of the analysis. Following the presentation of the basic functionality and the UX results, in Section 3.8, we perform a comparative evaluation with the current state of the art in the literature and commercial applications for blind navigation. In Section 4, we discuss the lessons learned, the various limitations of both the technical solution and of the Usability and UX evaluation process, as well as the future directions addressing these limitations. Finally, in Section 5, we briefly summarize the main points of the paper.

### 1.1. Background

Smartphones are now widely used by individuals of all ages and backgrounds. The main benefits of such solutions are the offered portability, adaptability, convenience, and user-friendliness [7,9]. This section categorizes the developed smartphone-based solutions targeting people that are blind and visually impaired.

Different types of sensors embedded on smartphones are utilized in [13,14,15,16,17,18,19,20,21,22,23,24,25,26,27] to record real-world environment data and, subsequently, employ the available smartphone processors to interpret the data and signal events.

Lin et al. [27] developed an assistive navigation system based on a smartphone application that can be used with an image recognition system. The system can operate in one of two modes: online or offline, depending on network availability. When the system is turned on, the smartphone takes a picture and transmits it to the server to be processed. To distinguish between distinct obstacles, the server employs deep learning algorithms [28,29]. The system’s primary drawbacks are its high-power consumption and the requirement for high-speed network access.

The TARSIUS system [30] was designed to help people better grasp visual scenes in outdoor settings. The TARSIUS mobile app, a web server, a remote support center and Bluetooth LE/iBeacon tags deployed along streets at sites of interest are all part of the system. However, the deployment of Bluetooth beacons all over the streets, which is expensive and may create signal interference, is one of the TARIUS system’s key problems.

ENVISION [31] employs a special approach to identify static and dynamic obstacles reliably and correctly from a real-time video stream captured by a smartphone with an average hardware capacity. If the obstacle identification and classification modules can assist the target users in gaining a better grasp of the environment, the system may be improved even further.

Lock, Cielniak and Bellotto [32] present a multimodal user interface that employs audio and vibration signalling to send navigational information to the target user. This effort is part of the “Active Vision with Human-in-the-Loop for the Visually Impaired” (ActiVis) project. The limitation is that it depends on Arcore to run, which is not supported by all smartphone devices.

The Tactile Wayfinder [33] is made up of a tactile belt and a Personal Digital Assistant (PDA) that runs a Wayfinder program. The app keeps track of the user’s position and path. The information is provided to the tactile display when the travel direction has been determined. The vibrators in the belt can provide information to the user about navigation directions. The problem in this application is the difficulty in the adoption from the users due to the external equipment required for its correct operation.

Tepelea et al. [34] describe a smartphone-based navigation system for blind people that makes use of the MEMS (Micro-Electro-Mechanical Systems) built into smartphones. Information to the user is communicated via a Text-to-Speech (TTS) interface and the app utilizes WiFi and Bluetooth to connect to external modules. The suggested portable system is effective, inexpensive and compact; however, it has not been tested in use cases including buildings or outdoors scenarios.

Alghamdi et al. [35] proposed a new approach for visually impaired and blind persons, to help them in interior and outdoor navigation by displaying their position and leading them to their destination. The system uses RFID technology that covers approximately a distance of 0.5 m. The test results demonstrate the accuracy of the suggested framework to be in the range of 1 to 2 m. However, the system’s accuracy detecting mechanism is not well-defined.

Tanveer et al. [36] developed a walking aid for the visually impaired based on a smartphone-enabled custom-made wearable device. When the obstacle’s position is recognized, the smartphone application creates Bengali/English speech signals. GPS is used to determine the latitude and longitude of the user, and the blind person’s position is tracked via a Google Maps-based application. The overall reported error rate is around 5% in the case of concrete and floor tiles. However, in some conditions, this method fails, with a notable example being spaces with floor elevation.

Patil et al. [37] propose a system (NavGuide) that creates a logical map of the surroundings to provide feedback to the user about obstacles, wet floors and ascending staircases. It consists of (1) six ultrasonic sensors, (2) a wet floor detector sensor, (3) a step-down button, (4) microcontroller circuits, (5) four vibration motors and (6) a battery for power supply. The system is mounted on the users’ shoes and vibration is the main way of interaction. The battery is reported to last, on average, around 600 min, but this is highly dependent on the used case scenario. Furthermore, the authors mention that the proposed solution is not heavy while the cost is low. They also demonstrate the effectiveness of the system in minimizing collisions with obstacles, especially when compared with the more traditional white cane. However, the system cannot detect pits or downhills, downstairs or wet floors prior to the user stepping on it.

Vidula V. Meshram et al. [38] propose a custom-made system mounted on a cane (NavCane) for both outdoor and indoor navigation that supports the following: (1) priority information about obstacles in the path without causing information overload, (2) obstacle detection at the foot, knee, waist and chest levels and scaffold objects up to the chest level and (3) object recognition in known indoor settings. The system consists of several components including: (1) five ultrasonic sensors, (2) a wet-floor detection sensor, (3) an accelerometer, (4) an RFID reader, (5) a contact button, (6) a vibration motor, (7) a GSM module, (8) a GPS module, (9) a single board small computer (SBSC) and (10) an external battery for the power supply. The user interacts with the system via audio signals and vibration motors. The reported battery consumption is, on average, over 600 min, the proposed system has a relative low cost while the added weight does not put a high burden on the user. The evaluation of the system involved 80 visually impaired people and demonstrated among others that collisions with outdoor objects were significantly reduced. Nonetheless, the system has the following limitations: (1) unable to recognize objects in unfamiliar indoor environments and in head-level obstacles, (2) the reported deviation is 4% between the actual and detected distance measurements of obstacles and, finally, (3) the identification of descending staircases as well as slopes is successful only if NavCane is held upright, forming approximately 90 angles with the floor plane.

Rahman et al. [39] present a system that works in unfamiliar environments and detects obstacles around the individuals’ left, front and right direction. It consists of (1) three infrared sensors, (2) a raspberry pi, (3) an external power supply and (4) headphones. The user interacts with the system via audio signals sent to the user’s headphones and a vibration motor. According to the authors, the cost is USD 45 and its net weight is 368 g. The evaluation of the system shows an average accuracy of 98.99% and error rate of 1.006%. Finally, the system is limited in terms of ground objects’ identification, while the authors do not present the power consumption requirements.

Chang Wan-Jung et al. [40] present a system that detects aerial obstacles and fall events on roads. When a fall event occurs, an urgent notification is sent to either family members or designated caregivers. The system consists of (1) wearable smart glasses, (2) an intelligent walking stick, (3) a mobile device app and (4) a cloud-based information management platform for sending the relevant notifications. The user interacts with the system with the help of vibration motors mounted on the cane. Experimentation with the system shows an average fall detection accuracy of up to 98.3%. However, the system cannot recognize front aerial and ground images such as traffic signs and traffic cones, while the authors do not disclose any information related to the power consumption requirements, the cost and the weight of their proposed solution.

Cardillo et al. [41] propose a system utilizing a microwave radar mounted on top of the traditional white cane that makes users aware of the presence of an obstacle in a wider and, thus, safer range. It consists solely of a microwave radar while the user receives feedback via acoustic warnings and vibration. According to the authors the system is cost-effective as it made of commercial components and lightweight, while there are no data on the power consumption required. Experimentation with the system shows a detection range of 0.5–3.5 m. Nonetheless, the system is still an early prototype and other limiting factors, such as size, have not yet been considered. 

Cardillo et al. [42] present a system that warns the user about the presence of humans in complex environments with the concurrent presence of multiple moving targets. The main method of detecting human obstacles is a novel range alignment procedure that detects the chest displacement. It consists of a white cane with a mm-wave radar mounted on top of it. The user receives feedback via an acoustic and/or haptic interface. The weight of the proposed solution is estimated to be low while there are no data from the authors on the power consumption requirements or the cost. The effectiveness of the proposed solution is validated via both simulated and experimental results. A weakness of the system is its reduced effectiveness for stationary human targets.

Kiuru et al. [43] present a system based on a frequency-modulated continuous wave (FMCW) radar principle where it detects obstacles in a 25-degree horizontal angle, covering a 0.9 m-wide area at a distance of 2 m in front of the user. The system’s vertical angle is approximately 70 degrees, making it possible to detect obstacles 1.4 m above the position of the sensor at a 2 m distance. The radar-based device is worn as a heart rate monitor and the user receives feedback via sound or vibration. The range from which the prototype vibrates or issues voice instructions is set to 3.5 m. The authors report that the prototype is light and the battery lasts for up to 4 or 5 h max when full functionality is on. However, there is no information regarding the cost of the proposed solution. On the downside, the authors do not perform an experimental evaluation of the system besides a user satisfaction study claiming an overall score of 3.8 out of 5. Furthermore, we can conclude that the battery is not ready for everyday use while, from our experience, not all blind individuals are particularly happy with a solution that demands a wearable device to be attached around their chest.

Islam et al. [44] propose a walking guide system that detects obstacles in three directions (front, left and right) and road surface potholes using an ultrasonic sensor combined with a convolutional neural network (CNN). It consists of (1) ultrasonic sensors, (2) a single Board Computing (Raspberry PI), (3) an RPI camera, (4) headphones and (5) an external power supply. The system is mounted on the user’s head, receiving feedback via audio signals. According to the authors, the proposed solution costs USD 140 and weighs around 360 g. Information about the power consumption requirements is not provided. The experimental results report an accuracy of 98.73% for the front sensor with an error rate of 1.26% (obstacle 50 cm distance), while image classification achieves accuracy, precision and recall of 92.67%, 92.33% and 93%, respectively. However, the system’s requirement of headphones raises issues to the blind and visually impaired as it could potentially cover ambient sounds critical for their safety. Furthermore, even though the objects and potholes are detected, the system cannot categorize them and, finally, the system’s size and weight raise questions about its wearability.

Elmannai W. M. et al. [45] present a system to avoid front obstacles utilizing sensor-based and computer vision-based techniques, as well as image depth information and fuzzy logic. It consists of (1) a FEZ spider microcontroller, (2) two camera modules, (3) a compass module, (4) a GPS module, (5) a gyroscope module, (6) a music (audio output) module, (7) a microphone module and (8) a wi-fi module. The user receives feedback via audio signals. According to the authors, the system costs USD 242.41 and weighs 180 g. Information about the power consumption is not presented. Experimentation with the system shows an achieved accuracy on detecting objects of 98% and 100% accuracy in avoiding them. Nonetheless, the system may not be able to detect either walls or large doors due to the size of their representation on the image and, finally, it is not the most cost-effective solution.

Duh, P. et al. [46] propose a system based on a novel global localization method (VB-GPS) and image-segmentation techniques with a single camera for a better scene understanding of detecting and warning about moving obstacles, providing the correct orientation in real time or supporting navigation between indoor and outdoor spaces. It consists of (1) two servers, one for MBL and the other for semantic segmentation, (2) a local computer, (3) a wearable camera and (4) a smartphone. The latter is used by the user to receive feedback via audio signals. The experimentation section demonstrates the precise locations and orientation information (with a median error of approximately 0.27 m and 0.95), the ability to detect unpredictable obstacles and to support navigation in indoor and outdoor environments. The authors do not provide any information about the energy requirements, cost or weight of their solution. Recognized weaknesses of the system include (1) its reduced reliability due to its sensitivity to network communication delays, (2) poor localization and scene-understanding results in rainy days or at night and (3) prepared information about static objects in advance via a landmark map.

Lin et al. [47] present a deep-learning-based assistive system with an obstacle avoidance engine that learns from an RGBD camera, semantic maps and pilot’s choice-of-action input. It consists of (1) a smartphone, (2) earphones, (3) a stereo based RGBD camera, (4) a wearable terminal with sunglasses and (5) an external PC. The system provides a voice interface to the user, it weighs no more than 150 g and achieves an accuracy of 98.7% in daylight conditions and 97.9% at night. The authors do not include any information at all about the power consumption requirements and the cost. Among the weakness of the system are its susceptibility to different lighting scenarios and its form factor impacting wearability.

Younis et al. [48] propose a new context-aware hybrid hazard classification assistive technology to help bring attention to possible obstructions or hazards to people with peripheral vision loss. The system provides capabilities such as hazard detection recognition, hazard tracking and real-time hazard classification modules. It consists of computer-enabled smart glasses equipped with a wide-angle camera and a MacBook laptop. The user interacts via visual and audio signals. Experimentation with the system reveals a 90% True Positive Rate (TPR), 7% False Positive Rate (FPR), 13% False Negative Rate (FNR) and an average testing Mean Square Error (MSE) of 8.8% on both publicly available and private datasets. The authors present no information about the power consumption requirements, the cost or weight of the proposed solution. Among the weaknesses of the system are the fact that it is an early prototype and the limited personalization for the notification style.

Yang et al. [49] present a unified approach based on seizing pixel-wise semantic segmentation providing qualified accuracy while maintaining real time speed and reduced latency over vision-based technologies with monocular detectors or depth sensors. It consists of (1) smart glasses, (2) an RGB-D sensor (RealSense R200) and (3) a set of bone-conduction earphones. The system provides real-time acoustic feedback by synthesizing stereo sounds (clarinet sound). The experimentation section reports an accuracy of 96% in the context of traversable area parsing using the real-world TerrainAwarenessDataset, outperforming other state of the art solutions. The authors do not disclose information about the power consumption requirements, cost or weight of their proposed solution. A weakness of the system concerns a lack in perceiving crosswalks and traffic lights, hazardous curbs and water puddles.

Bai et al. [50] propose a wearable assistive device that allows navigation in unfamiliar environments, as well as object detection and object recognition based on a lightweight Convolutional Neural Network (CNN). It consists of (1) a Red, Green, Blue and Depth (RGB-D) camera, (2) an Inertial Measurement Unit (IMU) mounted on a pair of eyeglasses and (3) a smartphone. The system utilizes an audio module for user feedback that emits a beeping sound for obstacle alert, uses speech recognition for user commands and uses speech synthesis for conveying information about the environment. The experimentation with the system demonstrates a decrease of the time required for navigating on the order of 5–10% and an 87% reduction in obstacle collision. The system’s cost is relatively high and has a medium weight, however, there is no information on the power consumption requirements. Among the limitations of the system include the inability to detect small-sized obstacles and staircases, while there is no tactile feedback. Finally, the proposed solution is an early prototype.

Long et al. [51] present a fusion system for perceiving and avoiding obstacles. It consists of a millimeter wave radar and RGB-depth sensors, while it provides a stereophonic interface to the user. The experimentation with the system reveals an expansion of the effective detection range up to 80 m compared to using only the RGB-D sensor. Nonetheless, the proposed solution is bulky and has a high cost. Furthermore, the system is limited to detecting objects and not recognizing them, while it is not portable as it still runs on a PC.

Cheng et al. [52] propose a system for crossings that uses RGB-Depth images to inform the user about the crosswalk position (where to cross roads), crossing light signals (when to cross roads) and pedestrian state (whether it is safe to cross roads). It is able to detect multiple targets at urban intersections. The utilization of RGB-Depth images allows for both increasing detection precision when compared with plain RGB images and to convey the distance of the detected objects. The system consists of wearable smart glasses and provides user feedback via a voice interface. The cost of the proposed solution is relatively high while the weight is estimated to be low, and the authors do not include any information about the power consumption requirements. The system is evaluated on real scenarios with success, while the reported time for processing a frame is at 200 ms. Among the limitations of the proposed solution are the inability of operating in night scenarios. Finally, a more elaborate experimentation section would allow for a more extensive evaluation of the system’s capabilities. 

Yu S. et al. [53] present a system aiding crossing intersections by utilizing a modified convolutional neural network version of LytNet based on MobileNetV3. The latter runs on a smartphone device and feedback is provided to the user via auditory and vibration signals. According to the authors, the DNN model achieves a classification accuracy of 96%, an average angle error of 6.15 while running at a frame rate of 16.34 frames per second. The cost and weight are directly related to the smartphone device of choice, while there is no information regarding the power consumption requirements. Among the limitations of the system are the lack of support for crossing intersections at night, being an early prototype and region locked, as well as the specific orientation and position requirements of the smartphone device.

Ihejimba et al. [54] propose a highly available, highly scalable, low-latency IoT edge computing solution for traffic light notification. It consists of (1) a Raspberry Pi 4 (Linux), (2) a smartphone device and (3) access to the AWS cloud. The user receives feedback via voice instructions and vibrations. The evaluation of the system demonstrates the average response time to be around 19.2 ms, the lowest response time to be 10.22 ms and peak response time to be 36.05 ms. The cost and weight of the proposed solution is moderate and light, respectively, while there is no information on the power consumption requirements. Nonetheless, since the system is Cloud based, it cannot truly guarantee low-latency and it is sensitive to connectivity issues. Furthermore, the system has no accurate and precise knowledge of the user’s GPS location, and it is not capable of local offline decisions.

For a more extensive overview of the available solutions and policies related to smart traffic lights and crossing intersections worldwide, it is recommended for the reader to refer to the study of Theodorou et al. [55].

Saez, Y et al. [56] present a system that assists mobility in public transportation based on RF communicating modules. It allows one to request a bus service by giving information to bus drivers, boarding the correct bus and reaching the destination easily and safely. The system consists of three modules: (1) MOVI-ETA, (2) MOVI-STOP and (3) MOVI-BUS. Specifically, the MOVI-ETA module includes an ATmega328P microcontroller (eight bits, sixteen MHz, AVR architecture) and an HC-12 wireless serial port communication device. The MOVI-STOP module includes two microcontrollers, two TI-CC1101 RF transceivers and an HC-12 device. Last but not least, the MOVI-BUS module includes two microcontrollers and two TI-CC1101 RF transceivers. The user holds the MOVI-ETA module and interacts with it via audio and vibration signals. According to the authors, the proposed system has a low cost and is estimated to be lightweight. However, they do not provide information regarding the power consumption requirements. Various field tests demonstrate the feasibility of the proposed solution. Among the weaknesses of the system are (1) the requirement of holding an extra custom-made device, (2) the changes and extensions to the infrastructure regarding stops and buses, (3) the communication range, (4) the average number of transmission errors, (5) the placement-dependent signal power reception, (6) the reliability that depends on the specific conditions of the environment, (7) the lack of multi users and multi buses arriving at stops in the test scenarios and, finally, (8) the lack of usability studies are considered. 

Yu, C et al. [57], propose a system that provides a seamless bus reservation service with minimal notification by utilizing the available bus telematics system and tactile surface indicators at bus stops. It utilizes a smartphone device while the user can interact with the service via auditory signals in the case of blind individuals, as well as via a GUI for people with low vision. The system also utilizes vibration motors for providing feedback. According to the authors, the cost and weight is low and light, respectively, while they do not include information concerning the power consumption requirements. The experimentation with the proposed solution is limited as neither metrics were presented, nor large field tests conducted. Furthermore, although it minimally disrupts the bus driver, it still requires cooperation.

See A. R. et al. [58] presents a system that enables object and obstacle detection in a single app. It only requires a smartphone with a single depth camera. The user interacts and receives feedback from the application via voice, audio, gestures and vibration. The cost of the proposed solution is relatively low and is lightweight, however, the authors do not provide information about the power consumption requirements. The experimental results demonstrate the ability of the system to detect outdoor objects at a distance of 1.6 m. Among the limitations of the solution are the inability to identify the name and type of the detected object, as well as the number of detected objects being currently limited to the ones with which the system is trained.

Last but not least, Meliones A. et al. [59] proposed an obstacle detection algorithm as a component of a mobile application that analyzes, in real time, the data received by an external sonar device. Its main function is to detect the existence of obstacles in the path of the user and to emit information, through a voice interface, about the located distance, size and the potential motion and to advise as well how the user can avoid them. The proposed system consists of a smartphone, and an external device. The latter is comprised of the Atmega 328p microcontroller, the U-blox NEO-8M GPS receiver, the HC-SR04 Ultrasonic sensor and the MG90S Micro-servo motor. The user interacts with the device and receives feedback via voice instructions. The proposed solution is cost effective (EUR 60) and lightweight. The experimentation results demonstrate extensively the effect of the proposed solution on the CPU, memory load and battery life. Furthermore, a number of real-life scenarios with different types of obstacles and the generated results demonstrate the effectiveness of the proposed detection algorithm. Overall, the proposed system shows reliable and robust results even when using cost-effective wider ultrasonic beam sensors in the context of a sparse city environment with wide pavements. However, in the context of a denser city-like environment, the cost-effective sensors demonstrate poorer results, mainly due to their reduced directionality. This limitation can be alleviated by integrating the existing outdoor blind navigation framework with narrow/pencil beam ultrasonic sensors that can produce efficient results in this context as well. However, this happens at the expense of significantly increasing the cost, signifying a non-optimal solution of the proposed system. The object detection system as well as the entire outdoor navigation system, the evaluation of which is the scope of this paper, is described in the following section in more details.

### 1.2. System Description

The proposed implementation aims at supporting people with limited vision or complete blindness to navigate with high precision and safety in outdoor spaces without the aid of guides. The system provides continuous feedback to the blind person containing critical information, via issuing voice instructions, that pertains to ensuring the correct and safe navigation of the individual as well as detecting obstacles and, subsequently, providing guidance on how to avoid them. The system is comprised of two subsystems that are tightly integrated. These include a wearable device incorporating an external GPS receiver with high precision tracking pedestrian mobility in real-time, a second device with an ultrasound sensor mounted on a servo mechanism functioning similarly to a sonar, an Android application that acts as the central component of the system and, finally, an appropriate (custom-made) voice interface to enable fast and accurate user interaction with the application. The user, via the application’s voice interface, can select the desired destination. As soon as the destination has been selected, the navigation process starts. To provide robust navigation information for blind individuals, the application receives data from both the Google Maps service and the Athens Public Bus Transportation (OASA) real-time telematics service that includes timetables and stops of urban transport services. The data are, subsequently, fed into a novel routing algorithm that provides high-precision navigation coupled with the ability to configure complex routes that may include Public Means of Transport mobility.

A carefully designed set of voice instructions provides the required information to ensure the correct and safe navigation of the users, as well as to convey information about potential obstacles along their path. The voice system interaction (instructions, information and options requesting user response) is better experienced via the use of bone conduction headphones, not suppressing the environment sounds which is critical for enhanced perception and safety in blind outdoor navigation. In general, the role of the application will be supportive of the users’ actions and will prioritize their safety as it will include the possibility to make emergency calls. Figure 1 shows the high-level architecture of the proposed system.

#### 1.2.1. Design Process

For the conceptualization and the implementation of the prototype, we followed a cognitively informed design process [60]. This approach integrates the usual iterative process of the engineering method with cognitive factors and processes related to how individuals attempt to solve their problems systematically. Moreover, this cognitive design framework promotes a set of principles-criteria that include safety, reliability, reinforcement and preferences, and emphasizes the inclusion of the immediate beneficiaries in the design process as well. In our case, the developed solution addresses the problems of navigating blind and visually impaired individuals in outdoor spaces.

The first step of this participatory design is to create an understanding of users’ needs [61]. Thus, we start with the needs assessment and, since the broader social context affects these needs, we also have to take into consideration the social constraints. We assume that individuals who are blind have similar skills as do the sighted, but that the richness of their environmental information is severely hampered [61]. Conducting interviews with the disabled population helped to unearth the cognitive processes involved during navigation, a requirement of this design approach, for enabling the functional needs assessment. The benefits of these activities are all about making the practices explicit, including stated preferences, habits and psychological features of blind and visually impaired individuals with respect to the use of mobile technology [61], as well as the multiple psychological constructs, such as interest, focus, enjoyment [62,63] and the benefit of usage [64]. For example, visually impaired users prefer routes that are often not the shortest ones, but are based on users’ proficiency and preferences [65]. The above, besides providing a basis for understanding the required functionality to support, also helps in engaging the user and potentially increasing the perceived quality of user experience, which is also a critical goal of the proposed solution. Furthermore, this will help with another important aspect of the proposed navigation application which is for the content of the issued instructions to be of high quality by leveraging the users’ situational context to reduce navigation errors.

Last but not least, we considered all the useful recommendations given in [7] during the design and implementation phase. These include the following: an appropriate choice of real-time object detection methods, mitigation of the extensive learning time, comfortability in carriage and usage, the right amount of information given for safety reasons, the avoidance of social stigma, proper management and security of personal and private data.

#### 1.2.2. Subsystem Interaction

The navigation starts with the app prompting the user to provide input regarding the desired destination and whether to utilize any available means of public transport, in particular, public buses, since this is supported in the current version. After the user’s confirmation of the desired destination, the app proceeds to the stage of planning the navigation route and issues a high-level description of that. Simultaneously, the external device starts a loop in which it continuously collects data from the sonar and GPS sensors and sends them to the central Android application via Bluetooth, which in turn acts by modeling and analyzing them. In particular, the application, after receiving the data from the external device, starts two processes. The first is to navigate the user via leveraging the Google Maps service and to update its current list of the available Public Means of Transport to that destination. The second is to analyze the sonar and GPS data received from appropriate sensors integrated into the external device. The former is used to detect obstacles on the path of the user while the latter is used to report back to the user his position with a negligible margin of error (<1 m). In this way, the continuous flow of data received from the external wearable device allows for the system to adapt to the dynamically evolving environment. To better understand the interaction of the aforementioned components, the reader may refer to Figure 1. The MANTO project contribution involving reliable ultrasonic obstacle recognition for outdoor blind navigation is presented in detail in [59].

#### 1.2.3. Tracking—Navigation with Great Accuracy Exploration of the Application

This section presents the assessment, conducted during the pilot stage of the MANTO project, of the high accuracy navigation capabilities, the correction of the user’s routing and the repositioning back on the navigation path in case the user accidentally deviated from it. The figures are snapshots taken from the smartphone while the application is running and navigating the user to the destination during the pilot stage. Specifically, they depict the application’s information regarding the position of the user (red pin), the vector of the direction along the navigation path (green line) and the corresponding voice instruction (grey box). To accurately navigate the user when approaching corners, the application starts issuing the turn instructions more frequently by leveraging the higher tracking density of the external GPS receiver (see Section 1.2.5) and the functionality of the scheduler (see Section 1.2.4). When the user selects the desired destination, the navigation starts by providing an overview of the total route and an estimated time of arrival (see Figure 2).

Next, as the user moves along the navigational route, the application issues instructions guiding the user during straights and turns. In case of a mistake, the application will issue recovery instructions to place the user back on the correct navigation path. Figure 3 depicts the case where the user makes a mistake and instead of taking a right turn, a left turn is made placing the user’s vector in the opposite direction to the designated destination. When the app detects the error, it issues instructions based on the hands of the clock informing, in this example, that the correct direction is between 6 and 7 o’clock. After the user makes the correct adjustments, the application issues the instruction to move ahead in Grigoriou Lampraki street (Figure 3—right).

Finally, all the above are part of a trial route where the blind user, accompanied by members of the research team, started from the exterior space of the University of Piraeus at Deligiorgi 114 towards the departments’ laboratories at Odyssea Androutsou 150.

#### 1.2.4. Real-Time Voice Instruction Scheduler

This module is responsible for selecting the voice instructions to be issued by the Android device to the user. The goal is to, firstly, meet the real-time requirements and, secondly, to manage all the issued instructions and their emission frequency to the user. This is achieved by a completely fair scheduler that utilizes a Red-Black tree as its underlying data structure. The weight of each tree node is calculated by considering the priority and time order of the instructions as well as the number of attempts made to issue the latter. It is worth mentioning that great emphasis was given to the second part of the aforementioned goal as it was a major requirement highlighted by the analysis of the interviews. Specifically, the blind users requested for the application to offer the minimum amount of information at a reasonable rate to secure that the users will be the least distracted from the emitted sounds. Therefore, to avoid disturbing the user from consecutively issuing instruction notifications, we designed appropriate priority levels by distinguishing them into the following types:Corner: it concerns the case of an upcoming turn with the highest priority.NavigationFlowCritical: It concerns the case of an instruction of critical importance. It is mainly used for instructions that help users to recover from an error back to the correct navigational path, leveraging the vector of the user’s path. It is classified as an interrupt job.NavigationFlow: It concerns the case of non-critical instructions. It is commonly used for instructions that guide the user to continue without any change, for example, “Continue straight”.TransitFlow: it concerns the case of instructions relevant to the Public Means of Transport.Summary: It concerns the instructions relevant to informing the user of the major navigational events along the route to the destination. This information is the first instruction issued by the application before starting the navigation.

Finally, examples of the real-time scheduler’s operation are the following:
The voice instruction “the bus will arrive in 3 min” is not issued when the time has elapsed.The voice instruction “Continue straight on” is not issued after a critical change voice instruction that prompts the user to return to the correct navigational path due to deviating from the navigation vector.When the voice instruction remains the same for a long period, for example “Continue straight on” in the case of a long straight, the scheduler limits the frequency by which it is selected for emission in order to prevent the user from being overwhelmed by instructions void of utility.

#### 1.2.5. Great Accuracy and Tracking Density

The following section presents the higher location accuracy and density, as well as the significantly smaller error of the reported user position of the external GPS receiver in comparison to the one integrated into the smartphone. The figures below are excerpts from a trial during the pilot stage. 

Firstly, as it can be seen from Figure 4, the external GPS receiver, represented by the satellite icon, has better user location accuracy than the phone’s integrated GPS, represented by the phone icon. The proposed system can achieve centimeter position accuracy by utilizing, on one hand, three systems in parallel choosing between GPS/QZSS, GLONASS, GALILEO and BEIDU and, on the other hand, the large surface of the external GPS receiver antenna that cannot be integrated on smartphone devices. Their combination produces much better accuracy for the actual position of the user while the phone’s GPS falsely reports the user even being on top of buildings. Secondly, there is a significant difference in the density of the points reported from the two GPS receivers. Specifically, as it can be observed from the above figure: nine points of the external GPS receiver against three points of the phone’s GPS (Figure 4—left), thirteen points of the external GPS receiver against four points of the phone’s GPS (Figure 4—right). Thirdly, with the help of the small scale located at the bottom right, the difference in the error between the two GPS receivers is evident. There are cases where the reported user location given by the smartphone’s GPS receiver is 10 m away from what the external GPS receiver reports. Lastly, the error of the latter receiver is found to be less than 1 m.

#### 1.2.6. Navigation Route Combined with Public Means of Transport (Buses)

This section presents the case of selecting to include public buses as part of the navigation route. As a primary source of information about the location of the stops and the arrival time of the buses, the application utilizes the OASA Telematics service, a Greek public service that supports real-time information for buses. In the case that the latter service is unavailable, the application has designated the Google Maps service as a backup mechanism, which provides the required information, albeit, with reduced accuracy. The trials during the pilot evaluation phase of this feature’s functionality were conducted with the aid of blind subjects. Both the route and voice instructions issued from the moment of entering to the moment of exiting the bus were recorded. The following example is an excerpt from a trial in the field where the user chose Makrigianni Square in the district of Dafni as the destination and the nearest stop to Ymittos Square in the district of Ymittos as the starting point.

The figures below (Figure 5 and Figure 6) show the application’s functionality that issues notifications about the intermediate stops through which the user passes until reaching the destination. At each stop, shortly before arrival, a notification is issued and, unless it is the terminal stop, it alerts about the next stop.

Finally, when the blind user has reached the intended destination, the app alerts him/her to exit the bus (see Figure 7).

#### 1.2.7. Passing Traffic Lights Crossings with Safety

This section presents the case of passing traffic lights crossings with the aid of a second external waterproof (IP66) device mounted on every traffic light. The device sends to multiple Android smartphones, having BlindRouteVision installed, information about the status of the traffic lights (Green/Red) and the remaining time until the next status change occurs. The latter information, along with the information retrieved from a database, during the initialization of the application, includes the geographical position and other characteristics of traffic lights, such as the starting and ending point of the crossing, the direction of the vehicles and the number of the crossings, are used as inputs to help the user pass a crossing successfully. The pilot trial, the result of which is presented below, has been conducted on the marked crossing next to the traffic light of Doiranis and Athinas in Kallithea. The following figures are recorded from the smartphone device and present the user’s position, the vector of the user’s direction along the path and the corresponding voice instruction issued by the application. 

As can be seen in Figure 8 (left), when the user arrives at the crossing, the application detects the traffic light status and informs the blind user about the red status, prompting him/her to wait for five seconds until the status changes to green. When the level of the traffic light becomes green, the application informs the user of the time remaining to pass the crossing with safety (Figure 8—right).

During the execution of the trial routes, the trainer’s responsibility was to observe the procedure and to provide continuously constructive feedback. The following route is an excerpt of the available routes for training. In particular, Figure 8 depicts the blind user traversing outside the Lighthouse for the Blind of Greece in the region of Kallithea. The route starts and ends at the entrance of the Lighthouse for the Blind of Greece, Athinas 17. This procedure (trial) is repeated several times depending on the user’s capability and perception of the environment. 

## 2. Materials and Methods

### 2.1. Usability—User Experience (UX) Methodology

One of the primary targets of this paper is to quantify the Usability and User Experience (UX) of the application and validate the system design in those terms. The reason for evaluating these two measures can be attributed to the great need to know the extent to which a system can be easily learned, its usage efficiency, the error rate, as well as the degree to which a user can swiftly recall how to use it [66]. Towards that direction, we reviewed the literature on the available methods to assess them. According to ISO/IEC 25,010 2011 [67], usability is defined as “the degree to which a product or system can be used by specified users to achieve specified goals with effectiveness, efficiency and satisfaction in a specified context of use”. In particular, the 3 main components measure the following:Effectiveness—measures the degree to which users can complete a task.Efficiency—measures the time it takes users to complete a task.Satisfaction—measures, subjectively, the quality of interaction with the application.

On the other hand, UX is a term that is broadly used by many researchers and practitioners to include different concepts [68]. Thus, it is an umbrella term for a range of dynamic concepts, such as traditional usability, and it also includes affective, emotional, hedonic, experiential, and esthetic dimensions. According to ISO 9241-210:2019 [69], UX includes users’ emotions, beliefs, physical and psychological responses and it is also the result of brand image, presentation, system performance, the user’s internal and physical state resulting from prior experiences, attitudes, skills and personality, among others.

Therefore, instead of using more typical usability questionnaires such as System Usability Scale (SUS), Post-Study System Usability Questionnaire (PSSUQ) and others, we opted for UX questionnaires since this concept is more general and captures usability more broadly [70] as it includes other aspects such as user perceptions and responses both from the use and anticipated use of a product [69]. These characteristics of UX and the ones described above make clearer the fact that it is the appropriate measure for assessing the component of satisfaction as described in the definition of usability in [67].

According to Dνaz-Oreiro et al. [71], there are three most used standardized questionnaires for user experience evaluation. In particular, these are AttrakDiff, UEQ and meCUE. This is also indicated in studies such as those given by Lallemand et al. [72], Baumgartner et al. [73], Forster et al. [74] and Klammer et al. [75]. The number of questions including AttrakDiff, UEQ and meCUE, as well as the scales they employ and the theoretical models on which they are based, are listed in [72]. The authors note that AttrakDiff has been the most popular option since it was firstly introduced in 2003, while UEQ surpassed it in 2017 and 2018. On the other hand, with meCUE being a relative newcomer, it has a substantially smaller usage rate.

These approaches are frequently supplemented with others and, according to Dvaz-Oreiro et al. [71], over 60% of the cases utilized between one and five additional methods. For example, a widely used standardized usability questionnaire is the SUS that is found both in academia and in business. Hedonistic and pragmatic aspects of UX are found both on AttrakDiff [76], meCUE [77] and UEQ+. While meCUE considers the emotional component as well, thus providing greater insight into acceptance-related problems, its usage rate is comparatively lower, diminishing the value of the results when it comes to reaching safe conclusions.

### 2.2. Usability—Metrics for Effectiveness and Efficiency

The user study aimed to evaluate the usability of the developed prototype both for blind and visually impaired people, in terms of effectiveness and efficiency, and receive feedback from experts involved in this study. In search of finding appropriate statistical metrics for measuring effectiveness, we conducted a literature review where we found that among the most common ones are the following: Completion, Errors and Error Rate. Their simplicity makes it easy to understand them and, thus, they are widely used in many studies. Completion rate counts, either as a pure number or as a percentage, the successfully completed tasks while Errors count the errors made by a user, as its name suggests. Error rate reports the number of errors per user. Common cases of errors include, among others, mental errors, for example, when a user cannot comprehend a system option [78], and undesired results as a consequence of either poor interaction with the system’s interface or lack of the provided information resolution.

For the evaluation of completion, error and error rates, the research team defined three tasks:First task: completion of a pedestrian navigation route—this task demands the user to successfully complete both known and unknown itineraries that do not include the use of public means of transport or passing traffic lights crossings.Second task: combining pedestrian navigation with public means of transport (entering and exiting the bus)—this task demands that the user successfully complete the following steps:
arriving at the bus stopleveraging the information emitted for being aware when the bus arrivesentering the busactivating the public means of transport mode of operation by pressing the given interface elementexiting the bus at the correct stopThird task: passing marked crossings near traffic lights—this task demands from the user to pass a traffic light marked crossing in time.

For each of the above tasks, the definition of completion and error is defined as follows:First task:
Completion: successful termination of the navigation despite any errors made during the trial.Error: the user giving up the attempt or asking for help from the research team counts as an error.Second task:
Completion: successfully completing the steps described above.Error: if one of the steps described above is completed with the help of the research team or the user gives up, then it counts as an error.Third task:
Completion: successfully passing a marked crossing close to a traffic light.Error: if the user gives up or asks for assistance from a member of the research team, then it counts as an error.

Finally, the completion rate is calculated by the following equation: (1)$$Effectiveness=\;\frac{total\;\#\;of\;tasks\;successfully\;completed}{total\;\#\;of\;tasks\;undertaken}\;=\frac{\sum_{l=1}^{U}\sum_{i=1}^{Μ}task_{li}}{U*M}$$
where *U* = # of participants, *M* = # of tasks per participant and $task_{li}=i-th\;task\;of\;the\;l-th\;user$.

$\mathrm{Furthermore},\;task_{li}$ takes the value 1 if the task is successfully completed and 0 otherwise.

Efficiency is closely related to effectiveness as it considers the time (in seconds and/or minutes) involved in successfully completing a task. A common way to measure effectiveness is with the help of the following formula:(2)$$Efficiency=\frac{\sum_{j=1}^{U}\sum_{i=1}^{M}tasks_{ij}t_{ij\;}}{\sum_{j=1}^{U}\sum_{i=1}^{M}t_{ij\;}}\times 100\%\;$$
where $t_{ij}\;$= $EndTime_{ij}-StartTime_{ij}$, for which, in turn, *EndTime_ij_* is defined as the time required for the i−th task of the j−th user to be completed successfully or the time until the user quits.

Errors will be measured as simply the sum of each participant’s total number of errors:(3)$$Error=\sum_{i=1}^{N}e_{i}\;$$
where *N* = the total number of the participants, while the error rate is calculated by the following equation:(4)$$Error\;rate=\frac{Error\;}{P}$$
where Error=# of total errors and *P* = # of total participants.

Last but not least, an added benefit of the above metrics is their deployment flexibility as their required input can be collected during any stage of development.

### 2.3. UEQ+ Standardized Questionnaire

To the best of our knowledge, there are no questionnaires available that evaluate the user experience of blind and visually impaired individuals. One of the drawbacks of the existing questionnaires is the lack of specificity in the features they assess. In response to this inflexibility, the UX questionnaire framework UEQ+ was selected.

The latter is a modular extension of the very well-known user experience assessment tool UEQ that provides results consisting of easily processed quantitative data and which has proven to be appropriate in evaluating new technologies regardless of gender, age group, level of education and technological proficiency [79]. Specifically, the UEQ+ is a set of scales that are combined to form a concrete UX questionnaire. Therefore, it is possible to create a questionnaire that is custom made to fit exactly the features of the proposed application which is under evaluation. Each scale is decomposed into 4 items that measure the impression of the user towards the UX aspect under consideration and a single item that measures the relevance or importance of the scale for the user. The items are scored on a seven-point Likert psychometric scale. The rating is configured to look at opposite pairs of the app’s properties. Users, according to the evaluation instructions, always choose an answer, even when they are not sure about the evaluation of a pair of terms or even when they think that it does not relate to the product perfectly. Finally, the user states how important each scale is for the overall impression of the product. 

The UEQ+ framework currently offers several UX scales. After careful consideration of the challenges, of the results of the user requirements elicitation stage [61,80], by which the blind and visually impaired users become part of the process of defining the evaluation criteria, and of the importance ratings given by the blind users to a preselected set of scales, we concluded that the following scales best describe the UX impression for the features that are deemed to have a higher value. Based on the available categories of UEQ+, the following were selected particularly for our case:Efficiency: this scale evaluates the user’s subjective impression as to whether he/she must put in the minimum effort required to achieve the desired goal as well as how quickly the application reacts to the user’s actions.Perspicuity: this scale evaluates the ease with which users become familiar with the application and how easily they can learn it (educability).Dependability: This scale evaluates the subjective impression that the user has on the predictability and consistency of the system’s response concerning its instructions and actions. In other words, it examines whether the user controls the interaction with the application.Adaptability: this scale evaluates whether the application can be adapted to the personal preferences of the user as well as how easily and quickly this adjustment process is done.Usefulness: this scale evaluates the advantages that the user perceives in terms of achieving his goals, how much time he/she saves and whether it increases his/her efficiency.Trustworthiness of content: this scale evaluates whether the content of the instructions provided by the application is of good quality and reliable.Response behavior: this scale evaluates whether the response from the voice assistant is friendly and pleasant to the user.

### 2.4. Semi-Structured Questionnaires

We have designed a seven-point Likert scale questionnaire similar in scope to the UEQ and UEQ+. The format of semi-structured interviews was preferred over other options due to its flexibility, and despite the limitations that affect statistical analysis, it ensures that the views of the blind and visually impaired are underlined. Undoubtedly, this feature is one of the key factors of this technique’s success as the increased flexibility provides the required degrees of freedom in the design and refinement phase of the interviews’ questions. Simultaneously, this type of interview allows for both the research team and the interviewees to further clarify their thinking about the challenges and the desired functionality. On the other hand, an unforeseen implication that pleasantly surprised us was the eagerness and willingness of some of the participants to engage in the refinement process of the interviews, expressing at the same time a depth of feeling about the issues that were raised. In this way, the blind and visually impaired get to express their views and expertise as well as provide feedback both on how to better formulate the interview questions in greater depth and to highlight areas of improvement for the application’s supplied functionality. Finally, during the semi-structured interviews, a goal of high priority was for the participants to remain impartial to the interviewers’ expectations and create a safe environment where the participants could express their opinion openly without the fear of being criticized.

### 2.5. Description of the Evaluation Setup and the Interview Participants

This section presents the evaluation setup of both Usability and UX, as well as their respective results. In more detail, it will present the findings of the application’s usability, measured in terms of effectiveness and efficiency, along with the results from the application’s UX evaluation, assessing user satisfaction, combined with the lessons learned from semi-structured interviews to better understand the user experience evaluation score. The trials were conducted at the premises of the Lighthouse for the Blind of Greece, which is the main non-profit organization for the education and assistance of the blind and visually impaired in Athens. Specifically, the pilot phase was instrumented with the following data collection tools.

#### 2.5.1. Personal Characteristics Questionnaire

This questionnaire, filled in by the interviewees, aimed at learning various personal characteristics of the participants. In more detail, they were asked to fill in details related to their gender, age, degree of visual impairment, cause of vision loss and digital sophistication.

#### 2.5.2. Exploration of the Application

The functionality of the application’s characteristics was presented in the Orientation and Mobility (O&M) lessons with the aid of instructors. In this way, the interaction between instructors and trainees is encouraged, thus expediting the learning process via a more personalized and elaborate exchange of information. As a result, the benefits of this phase are twofold. At the same time, trainees ask freely for clarifications without having any feelings of inadequacy and instructors receive feedback from the trainees assessing both the comprehension progress and the rising opportunities that could potentially improve the learning process in the future.

#### 2.5.3. Pedestrian Navigation Tasks

Following the exploration phase of the application’s functionality, trainees were asked to perform pedestrian navigation tasks (see Section 2.2). These include both known and unknown routes that also incorporate public transportation (buses) and passing marked crossings near traffic lights, the details of which are described in Section 1.2.7. These tasks also include the utilization of the obstacle detection and avoidance subsystem that is based on the external sonar-like device. 

#### 2.5.4. Usability, UEQ+ and Semi-Structured Interviews

After the tasks were performed, the research team proceeded with the user evaluation aspect of the study. The research team measured effectiveness and efficiency (see Section 2.2), the UEQ+ questionnaire was used to assess user experience (see Section 2.3) and semi-structured interviews (see Section 2.4) were held for assessing functionality using both prearranged questions and encouraging the interviewees to share anything they thought was relevant. All the above data collection tools were used to find potential areas of improvement to further refine the functionality of the application. The format of the semi-structured interviews was particularly helpful in that direction.

#### 2.5.5. Recording of Trials

Throughout the pilot stage, the research team diligently recorded, in video format, every phase described above to avoid misinterpretations and guarantee the validity of the results. The recordings were made after receiving permission respecting any privacy concerns raised by the trainees.

#### 2.5.6. Evaluation Process

For the evaluation process, 30 male and female members of the community of the blind and visually impaired participated by executing a number of tasks and filling out a set of questionnaires. The subjects were between the ages of 30 and 60 and they ranged from having severe to complete blindness due to various causes. Most of the interviewees had low digital sophistication, which highlighted the requirement to provide special training sessions customized to their needs. These sessions were held in the vicinity of the Blindhouse of Greece. Although we acknowledge that the number of participants is not representative enough and does not help to draw strong conclusions, the results of both Usability and UX evaluation can be used to make assertions about the behavior of the application. Finally, the sample that was provided is representative of the beneficiaries of the Blindhouse of Greece concerning age, gender, age of vision loss and ability to use digital devices.

To assess the Usability aspect of the application, the research team evaluated a few tasks (see Section 2.2) during O&M training sessions held at the BlindHouse of Greece. In particular, the users for the case of pedestrian navigation were tasked to execute two known and two unknown test routes. The known routes include a route that starts from the entrance of the Blindhouse of Greece and cycles back to it and another one that starts again from the entrance of the Blindhouse of Greece navigating toward the Stavros Niarchos Foundation Cultural Center (SNFCC). For the unknown routes, we met the users at the entrance of the Blindhouse of Greece, and we escorted them to two locations undisclosed to them (a local market store and the Onassis Cardiac Surgery Center) that were designated as the starting point with the task to return to the Blindhouse of Greece.

For the case of pedestrian navigation where a user selects the option to include Public Means of Transport, the users executed two test scenarios starting from the nearest bus stop and heading to Piraeus using the bus lines 040 and 229. Last but not least, in the case of passing traffic light crossings, the users conducted one trial on the traffic light of Doiranis and Athinas in Kallithea.

According to Section 2.5.4, the research team that conducted the UX evaluations used two UX questionnaires. The first questionnaire, distributed via Google Forms, followed the format of a standardized one. For a UX evaluation questionnaire to be considered standardized it must contain a constant number of questions found in the same order by the participants and answered independently. An important feature of the standardized questionnaires that makes their usage quite extended is their cost-effective nature and simplicity since all it takes is for the user to complete it after having experienced the product or service in question. Finally, another attractive feature of these UX questionnaires, is that they are considered dependable and valid [81]. The visually impaired users had the opportunity to complete the Google Form questionnaire either with the aid of the personnel at the Lighthouse for the Blind of Greece or at their own time and place. The second questionnaire, which followed a semi-structured interviews format, concerned also issues of UX. On average, the first questionnaire required 20 min while the semi-structured interviews required 30 min. The descriptive characteristics of the interviewees are presented in Table 1.

## 3. Results

### 3.1. Completion Rate during Evaluation Activities

Table 2 presents the number of successfully completed tasks during the evaluation phase per user for each case scenario. According to Equation (1) from Section 2.2, in order to calculate effectiveness, we need to first know the number of successfully completed tasks and the total number of tasks undertaken [78]. The total number of tasks consists of those tasks for which the result of execution was either a success or a failure, while the number of tasks successfully completed is the sum of each individual classification shown in the Table 2. Finally, by utilizing the table data, we calculate effectiveness as follows: Total # οf tasks completed successfully = # of tasks “Completion of a route” + # of tasks “entering and exiting the bus” + # of tasks “Passing crossing with traffic lights” = 162.Total # of tasks undertaken = #number of tasks per user * #of participants = 7 × 30 = 210.

Hence,
$$Effectiveness=\frac{162}{210}\;\times \;100\%=77.14\%$$

The complementary metric of the completion rate (failure rate) is calculated as FAILURE RATE = Total # of failed tasks/Total # of tasks undertaken, where Total # of failed tasks = Total # of tasks undertaken—Total # οf tasks completed successfully. Therefore, FAILURE RATE = (48/210) × 100% = 22.85%.

### 3.2. Errors—Error Rate during Evaluation Activities

In order to gain an understanding of the participants’ ability to navigate using the application, during the evaluation phase, the research team recorded the errors, both recoverable and unrecoverable, while the users made attempts to use the full functionality of the application. These are pedestrian navigation with or without the use of public means of transport and passing traffic light crossings. The identified errors for the exclusive pedestrian navigation case were classified as follows:Collision with obstacles: records the cases where users collided with an obstacle.Veering: Records the cases where users deviate from the designated path and veer off to one side or the other. This also pertains to the conventional methods of a white-cane and/or guide dogs.External factor: records the cases where users are affected by an external factor such as another person on the path or the application glitches.Missed turn: records the cases where users either react too early or too late and miss the correct turn.Over-turn: records the cases where users over-turned and missed the correct navigational path.Issued instructions: records the cases where users request further clarification about instructions from the research team.

Table 3 presents the number of navigation errors made during the execution of pedestrian navigation tasks following the above classifications after the users completed their training sessions. The column ‘Assisted’ contains the number of unrecoverable errors where users required external assistance from the research team or the instructor and, thus, failed to successfully complete the task at hand. This means that those tasks are not taken into consideration when estimating the completion rate metric. The rest of the columns present the errors made by the user; however, they were able to recover by themselves and, thus, contribute to the completion rate metric. In total, we identified 166 errors where the users recovered on their own, while 24 required external assistance. The most commonly occurring error (44 out of 166) was the users colliding with obstacles. Although we have managed to deliver a functional version of the obstacle detection system, there are still scenarios, such as balconies or short trees, that need to be improved in the future. The second most often occurring error (28 out of 166) was the inability to always recover from an overturn back to the correct navigational path due to different cognitive capabilities, shaped attitudes, beliefs and preferences. The overturns were mainly either the result of the irregularities of some building blocks and other parts of the trial routes or the result of any other external factor that leads to an over-turn. The latter event is not taken into consideration when determining the errors caused by external factors, hence, securing that those errors are not double counted.

It is worth mentioning that the most common error requesting assistance from the research team was from the category of obstacle collision and whenever the user had to deal with multiple obstacles in close proximity that had different shapes, awkward angles and were made of materials that could not be properly identified by the selected ultra-sonic sensor technology. For this case, the error rate (Equation (4)) is 166/30 = 5.53 recoverable errors per user on average.

Likewise, the identified errors for the case of utilizing the Public Means of Transport and for the case of passing traffic lights crossings were classified as follows:

Public Means of Transport
Boarding button: this category records the cases of activating the public means of transport mode of operation.Boarding on the bus: this category records the cases where the blind and the visually impaired user enters the bus.Exiting the bus: this category records the cases where the blind and visually impaired user exits at the correct bus stop.

Traffic Lights CrossingsVeering: this category records the cases where the blind and visually impaired user deviates from the straight line of the user’s vector path.Reaction time to the status change notifications: this category records the cases where the blind and visually impaired user reacts to the traffic lights status change notification.

Table 4 presents the number of errors made by the users for each category after they completed their training sessions both in the case of pedestrian navigation combined with public means of transport and in the case of passing traffic light crossings. Similar to the above case, the columns ‘Assisted’ for each category contain the number of unrecoverable errors where users required external assistance from the research team or the instructor and, thus, failed to successfully complete the task at hand. This means that those tasks are not taken into consideration when estimating the completion rate metric. The rest of the columns present the errors made by the user; however, they were able to recover by themselves and, thus, contribute to the completion rate metric. In total, for the case of pedestrian navigation combined with public means of transport, we identified 70 errors where the users recovered on their own, while 16 required external assistance. The most often occurring error (44 out of 70) concerned the users forgetting to activate the Public Means of Transport mode of operation. We plan to address this issue in the future by providing both haptic and audio feedback as a reminder to the user. For this case, the error rate (Equation (4)) is 70/30 = 2.33 recoverable errors per user on average. Likewise, for the case of passing traffic light crossings, in total, we identified thirty-one errors where the users recovered on their own, while nine required external assistance. Both classifications of errors for this case were equally identified (15 and 16 out of 31) by the research team. We plan to address this issue in the future by providing both haptic and audio feedback in order to correct the user. For this case, the error rate (Equation (4)) is 31/30 = 1.033 recoverable errors per user on average.

### 3.3. Efficiency

This section presents the results of the measured efficiency. In contrast to the previous presentation on effectiveness, not all cases are considered. Since efficiency takes the time a user needs to complete a task as the input, it seems reasonable to exclude the ones for which their time completion is dependent on external factors besides the user’s actions. This implies that both the tasks that include the use of Public Means of Transport and passing traffic light crossings will not be measured in terms of efficiency, thus leaving the pedestrian navigation as the only category that will be studied. Specifically, in the case of pedestrian navigation combined with Public Means of Transport, the task’s completion time depends on the bus’ time of arrival as well as the traffic conditions. Likewise, for the case of passing traffic light crossings, the task’s completion time is upper bounded by the time a traffic light is set to change its status.

In particular, the efficiency of pedestrian navigation is measured by utilizing Equation (2) on the successfully completed routes (see Section 2.5.6 for more details). The first known route was completed on an average time of ten minutes and a standard deviation of two minutes while the second known route had an average of five minutes and a standard deviation of one minute. The first unknown route was completed on an average time of five minutes and a standard deviation of three minutes while the second had an average time of ten minutes and a standard deviation of two minutes. Taking into consideration Equation (2) and the above data, we measured the efficiency to be 74%.

### 3.4. Questionnaires, Interviews and Group Discussions Findings (UEQ+)

This section presents the statistical results from the evaluation of the pilot phase. These include the mean value and standard deviation for every scale, the consistency of the questionnaire, the importance ratings for the selected scales and a Key Performance Indicator (KPI). The latter is provided by the UEQ+ tool and allows for the overall evaluation of the UX impression. For the calculation of the KPI, the UEQ+ tool collects each scale rating for four items and one rating for the overall importance of the scale. Next, the relative importance of the scale and the scale mean per participant are calculated. Finally, the KPI is then simply the mean over all participants.

### 3.5. Mean Value and Standard Deviation for Every Scale

This section’s scope is to present various statistics, including the mean value, for every scale of the questionnaire, as well as for each of the items in which they are decomposed, the corresponding standard deviation and, finally, the relevant confidence intervals. (Remark: The mean value ranges from −3 to +3 instead of 1 to 7. The latter transformation is a result of trying to conform with the range values of the initial version of the UEQ questionnaire.)

Table 5 summarizes the average value, the standard deviation and the computed 95% confidence interval level.

Subsequently, the above table is depicted also graphically in Figure 9 and Figure 10.

Next, the above scales are then analyzed into their constituent items, four per scale, in order to highlight, in greater detail, the above observations. The order of the exposition for the individual items mirrors the order the scales are presented. 

For the scale of Efficiency (1.36), users judged that the application was found to be primarily very organized (1.53), practical (1.33), fast (1.30) and that it was very efficient (1.27). Users stated that they do not have to perform unnecessary actions and they do not have to wait too long for the application to respond.

For the scale of Perspicuity (1.25), users found the app understandable (1.53), easy to learn (1.27), easy to use (1.13) and having a clear structure (1.10). Specifically, users were satisfied with the interface as it follows practices already known to blind people that are also compatible with the widely used TALKBACK service. Additionally, they reported that the available functions were well organized. Moreover, the majority stated that the training version, although it could be improved, created the necessary conditions for learning to use the application properly, firstly, by eliminating external distractions and, secondly, by minimizing users’ doubts about the risks that arise while navigating in a dynamic environment with obstacles. Another feature that makes it easier to use the app is that it does not add any further cognitive load to users. This is due to lifting the burden of having the user memorize detailed information about their pedestrian navigation, such as constructing cognitive maps of the entire route and tracking in real-time the current position in it. Finally, they noted that the user manual is quite explanatory and helps in learning how to use the application, especially if it is demonstrated by someone trained in this assistive technology. This is also true even for digitally sophisticated blind individuals which are approximately a third of the participants.

For the scale of Dependability (1.38), users found that the application’s functions were predictable (1.47), supported their navigation activities (1.43), met their expectations (1.27) and gave them a high sense of security (1.37). Specifically, they commented on the accuracy of the app and the precision of the instructions given that the app is in the pilot testing phase. Additionally, they were satisfied with the snap response of the application when the user made mistakes during the navigation as well as with the issued instructions that redirected the user back to the correct route. The functionality that integrates Public Means of Transport (buses) with pedestrian navigation was found satisfactory, while passing marked crossings near traffic lights was much safer according to users. Last but not least, users found the shake functionality that reminds them of their current position extremely helpful even if the user is inside a speedy bus.

For the scale of Adaptability, users stated that the application’s reading speed capability can be adjusted (0.97) according to their needs, through the utilization of the Talkback service, while at the same time they were somewhat satisfied with the provided flexibility (0.73) to choose between different styles of navigation instructions, either in a rectangular or clockwise style. The instructions issued in the former style follow a more discrete approach that exclusively includes perpendicular moves (right, left, behind, straight), while the latter includes instructions that are issued in degrees based on the hands of the clock. However, the users found the lack of a wider range of virtual assistants somewhat restricting as the application currently supports only Melissa, thus justifying the lower score on this feature. Finally, users found that adjusting the settings was neither difficult (1.1) nor slow to characterize as useless (1.2).

Specifically, from Figure 11, it is evident that for the scale of Usefulness (1.39), the application was found to be very helpful (1.70), beneficial (1.47), useful (1.37) and sufficiently rewarding (1.03) since it significantly facilitated the navigation of blind people outdoors. Furthermore, to this end, another factor of contribution was the fact that users found the ability to switch between different applications while running the outdoor navigation application (BlindRouteVision) particularly useful.

For the scale of the Trustworthiness of Content (1.34), users found that the app provides trustworthy voice guidance content (1.43). It was also useful (1.23), plausible (1.43) and accurate (1.27). Users stated that the voice instructions were tailored to the specific requirements and needs of the blind and visually impaired and their frequency of repetition was within reasonable limits facilitated by the custom-made scheduler. Users also stated that they were satisfied with the battery consumption and the level notification rate.

For the scale of the Response Behavior (1.18), users judged that the application’s response is produced at a relatively satisfactory rate and is pleasant (1.23), natural (1.20), likeable (1.10) and entertaining (1.20). Additionally, many users noted that the instructions could be improved despite being already satisfactorily natural.

### 3.6. Distribution of Responses by Scale

Next, Figure 12 depicts the distribution of responses for each scale. Given the above analysis, it is evident that the overall rating was positive, with the majority of the responses having a score above five. On average, 77.8% of the responses had a score of at least five, while the scales of Adaptability and Response Behavior scored the lowest.

Overall, user experience was positively evaluated by all participants, with the scale of Adaptability being the exception. This scale, which describes the capability of customization to the user’s personal preferences, received the lowest score (1.00). On the contrary, the scales of Usefulness (1.39) and Dependability (1.38) received the highest scores as users found that the app removes restrictions concerning pedestrian navigation, while, at the same time, the app’s operations were found to be reliable and predictable, respectively. The scales that are close enough are the scale of Efficiency (1.36), as users found that their goals can be achieved both quickly and efficiently, and the scale of Trustworthiness of Content (1.34) that emphasizes the quality of the information provided during navigation. The scale of Perspicuity (1.24) received a score that indicates there is room for improvement on how easy it is for the users to familiarize themselves with the application as well as to learn its operation, followed by the scale of Response Behavior (1.18) that shows the desire of users for somewhat better-quality characteristics regarding the app’s issued instructions. Finally, as already mentioned above, the UEQ+ tool provides a Key Performance Indicator (KPI) for the overall evaluation of the UX impression. It received a score of 1.48, which is considered a positive result given that the scale ranges between [−3, 3].

Figure 13 describes the importance ratings as given by the blind participants about the selected scales. 

### 3.7. Consistency of the Evaluation Categories

Finally, we used Cronbach’s alpha coefficient to determine the reliability of the results based on user responses. There is no generally accepted rule of thumb on how large the value of the coefficient should be, however, in practice, a value greater than 0.7 is sufficient to qualify the results as reliable. Specifically, the table below (Table 6) details the Cronbach’s coefficient values broken down by scale. The observed values indicate that the results are reliable. Figure 14 depicts Table 6 as a bar graph.

### 3.8. Comparative Evaluation

#### 3.8.1. Advancing the State-of-the-Art

In contrast to our approach, none of the state-of-the-art solutions (Section 1.1) underline the necessity of introducing specialized training courses as part of an evaluation framework for Usability and UX aimed at blind and visually impaired individuals, if they even consider assessing them at all. The benefit of this activity is that we can further solidify the validity of the findings. Our framework also emphasizes user participation into the process of defining the evaluation criteria by incorporating needs, beliefs, opinions, characteristics, personality and attitudes that were the result of the user requirements’ elicitation stage [61,80]. In addition, it includes user-centered courses incorporated in the existing O&M courses of the Lighthouse for the Blind of Greece with the cooperation of the instructors. Finally, with the aid of a custom-made training tool, the blind individuals were able to become familiar with the application in simulated conditions and scenarios, avoiding external hazards and guaranteeing their safety.

The major motivations for the proposed approach of obstacle recognition against the limitations of the related work is the sophistication of the proposed method regarding the detection of object size and near-field moving-object tracking and the subsequent short oral warnings for avoidance.

Our proposed solution for passing traffic lights crossings has the advantage of being non-invasive and not imposing the requirement of interaction with the traffic light management system. To guarantee the user’s safe crossing, the system, via a patent-pending algorithm allowing the connection of the application with the traffic light device, emits critical information with high accuracy, including the traffic light’s current status and its transitions, the time remaining to traverse the crossing as well as the directionality of the passing vehicles. It provides true low latency guarantees as it is not affected by network connectivity issues that cloud-based solutions must deal with. Furthermore, neither light nor weather conditions play a role as they do in computer-vision-based systems. In contrast to our short-range server system, other beacon-based solutions face the limitations of not being able to transmit accurate timing information in real-time, while they need to solve the problem of recognizing the ID of the receiving passive signal among a multitude of almost identical signals from multiple intersections.

Another advantage of the proposed solution in comparison to the existing state of the art is the high accuracy and density of the users’ tracked location, achieved by the combination of the external device with the application that enables accurate navigation. The application, also, provides:(a)accurate instructions for recovering back to the navigation path when the user diverges from it;(b)route selection confirmation;(c)navigation to bus stops, safely boarding them, upcoming bus stop notifications and instructions to exit the bus.

The power consumption of the proposed solution can be distinguished in two separate cases, one involving the sonar-based detection system and one without it. The experimental results show that the external device, having a 3500 mAh battery installed, can demonstrate constant operation exceeding 12 h before recharging in the most demanding navigation scenario involving sending dense high precision GPS data to the smartphone application every 1 s. For the case with the sonar module active, the system exceeds 4 h of operation before recharging is required, mainly due to the servomotor dominating the power budget of the external device. In fact, had we swapped the servomotor with two sensors placed at both sides of the central sensor, then we could have achieved more than a 90% reduction in power consumption. However, the servomotor mechanism is indispensable for the configuration of the system utilizing the narrow/pencil-beam sensors due to the smaller required rotation step. For more details regarding the power consumption as well the CPU utilization of the impulse noise filtering software, the reader is advised to refer to [59]. Although a comparison between the power consumption characteristics of the existing hardware solutions would be of great interest, nonetheless, the majority of papers do not include relevant experimentation sections.

Finally, not one of all these efforts is an integral part of a holistic modern state-of-the-art reliable high-precision wearable navigation system relying on a smartphone.

#### 3.8.2. Commercial Navigation Applications

The BlindSquare application is a proven solution that combines existing technologies to help blind and partially sighted people in their daily life. Specifically, it helps the user to perceive the surrounding environment with the help of voice instructions. It works solely on Apple-related devices. The BlindSquare app uses both the GPS sensor and the compass of the smartphone and gathers information about the surrounding environment from FourSquare. By utilizing unique filtering algorithms, the Blindsquare app can decide what information is most relevant and reports back to the user the results via high-quality speech synthesis.

Lazarillo is a specialized GPS application that integrates mobility tools for the blind. Using audio messages, Lazarillo informs about nearby places, streets, intersections and so on. The GPS sensor is used even if the app is in the background. This allows the user to continue using the app without actively using the screen, even when the phone is in a pocket or when other apps are used. The goal of the app is to help every blind person reach the selected destination simply by giving voice notifications of nearby locations, institutions and shops, thus allowing the user to interact more actively within the city.

InMoBS is an application that aims to keep users on the correct path, describe the surrounding area and inform them about the dangers they may face in it, as well as to support them during passing crossings. To support all the above, an appropriate network-based system was developed where the user interacted with the service either through an application for smartphones (“InMoBS mobile”) or, alternatively, through an application based on web technologies (“InMoBS Home”). With the support of the central service provider node, the above applications achieve the desired objective of ensuring the correct navigation. To ensure a higher degree of accuracy of the users’ reported location during external navigation, a high-precision external GPS receiver connected via Bluetooth is used, while, with the support of WiFi-capable devices placed at traffic lights, the integrity of users is ensured when passing crossings.

In comparison to the Blindsquare application, BlindRouteVision follows the same general principles and shares a common set of functionalities. Despite this overlap, BlindRouteVision is superior in the following features. First, the BlindRouteVision app has a higher accuracy and localization density during the user’s outdoor navigation, significantly outperforming Blindsquare, which uses the smartphone’s GPS receiver. Specifically, according to the data measurements, the accuracy of the GPS receivers integrated into smartphones is less than 10 m, which leads to inaccurately reported locations, while the use of an external high-precision receiver by BlindRouteVision helps to achieve an error of less than 1 m.

In addition to the high density and accuracy of tracking, BlindRouteVision features an innovative navigation algorithm that continuously tracks the user’s gait with great precision along the route, constantly knowing the distance from the route, and corrects the user in real-time when he/she deviates. In addition, the resolution of the reference points of the route path is at least three times denser (from two to five times) than the reference points used by the Google Maps navigator. For example, the Google Maps navigator will report “Move straight to 23 Thivon Avenue”, while the Blind RouteVision app utilizes the set of reference points yielding a more descriptive route: “Move along Ethnikis Antistaseos Street. Turn right on Thivon Avenue. Continue straight on 23 Thevon Avenue”. In this way, it ensures that the user’s actual path and the path planned by the application are in close proximity and parallel to each other. Therefore, as it is evident from the above, the real-time tracking of BlindRouteVision surpasses Blindsquare’s corresponding functionality.

Another point of differentiation is the BlindRouteVision algorithm’s change of direction instructions, which are extremely precise with an error of less than a meter, unlike the navigation instructions of classic navigators that do not have specialized instructions for the blind. For example, a classical navigator in the corresponding case would issue the instruction “in 600 m turn right” followed by the instruction “turn right”, e.g., 100 m before the turn, even though at that point there may be no turn or there may be a vertical road with an opposite direction of vehicle flow, as the driver can only turn right at the next alley. In addition, the Blind RouteVision application very effectively integrates Public Means of Transport.

Furthermore, it can guide the user very accurately to the bus stop, much more efficiently than other applications. It then gives them real-time information about the bus arrival time. Inside the bus, the user receives notifications about the next stops and when to get off. After the user gets off the bus, the application automatically continues the pedestrian navigation. Regarding crossings, the application knows precisely with zero latency the red–green status of all traffic lights of a traffic junction that the user is approaching and accurately selects the correct crossing. It also knows with zero latency the remaining time of the traffic lights status, the number of crossings and the direction of the vehicles at each crossing. The application guides the blind or the visually impaired user very precisely to pass the crossing safely. However, the correct operation of this feature depends on whether the traffic light has a second external device installed. Finally, the other applications do not support this feature.

## 4. Discussion

The successful completion of the evaluation phase marked the end of the MANTO project. Throughout this period, the team obtained valuable experience around issues concerning the navigation of the blind and visually impaired and how these translate to the design, implementation and validation process of relevant apps, as well as how to administrate such projects. Additionally, the lessons learned from this process, being of paramount importance, include the following:The importance of having a guiding application that allows blind users to complete all their activities.The necessity to adopt a design process that involves the blind and visually impaired users for enabling the development of an application where users can recognize the functionality of the cognitive processes used during their navigation.The necessity to design and implement an organized training framework for increasing the adoption and learning rate of the application.The importance of blending the design process of both the educational framework and the technical capabilities of the system to get a better and more robust result.

### 4.1. Technical Limitations and Future Work of BlindRouteVision

We recognize that a limitation of the original design is a lack of sufficiently distinguishing the specific needs of blind and low-vision users. Although it is the common case to conduct a detailed analysis of the needs of blind individuals, it has become clear from interacting with users at various stages of the development and pilot phases that blind and low-vision users have substantially different needs and have access to a set of different cognitive processes and experiences affecting, differently, the navigation requirements and the assistance that should be provided to them. A similar problem is identified for people with congenital and late-onset visual impairment.

During the pilot phase, it became evident that the current implementation of the external sonar device cannot distinguish obstacles with the desired degree of reliability when used in dense urban environments, while some transient false readings affecting the accuracy of obstacle detection need to be eliminated. The latter is due to the combination of the sensitivity of the ultrasonic sensor and the periodic change in its direction caused by the servo motor. In addition, the current form factor of the external device does not satisfy all users in terms of portability and a portion of them prefer to carry only the high-accuracy external GPS receiver due to its smaller size. In the near future, we will address the above limitations by evaluating the characteristics of a wider range of sensors. From our experience, we found that, in practice, their operation diverges both from the nominal viewing angle and the beam pattern listed on the accompanying specifications and even produces different results depending on the features of the object. Additionally, the existing noise removal filter will be further optimized in order to improve the application’s behavior in scenarios involving frequently-detected obstacles that are part of the blind route, such as walls or cars parked along the road. On the other hand, addressing the portability constraints will be the subject of longer-term research as factors not exclusively related to the technical dimensions of the problem need to be weighed. While several solutions have been proposed and evaluated by blind and visually impaired participants, no final solution has been accepted.

Concerning the external device mounted on traffic lights, some recognized limitations of the current implementation are, on one hand, the distance at which the application can detect the traffic lights and, on the other hand, the maximum supported number of concurrent users that can be connected to this device. Although the current number is satisfactory, based also on the evaluation, the aim is to increase this number.

Besides further improvements to existing features, part of the strategy to strengthen the value of the app as a blind person’s assistant is the incorporation of free travel functionality. By choosing this form of navigation, the user, without having to choose a specific destination, will be able to be informed about various points of interest such as shops, museums, retail stores and so on as they walk. This information is emitted in two approaches; either the user requests the information from the app (pull-based interaction) or the app gives the information to the user at its own pace (push-based interaction). Part of the future work will include the following:The app should provide additional feedback to the user to recover in the case of over-turning.Participants would like for the app to provide the capability to control how much information is given to them (push interaction). Furthermore, the participants requested for both push- and pull-based interactions to be adjusted in order to better match their personal style of preference.Search based on shop names and general categories—it will be possible to constrain the search results returned to the user based on the selected coverage radius. It will also be possible to search either via shop names or via more general categories.Integration of the application with social networks—instant connection and presentation of news related to points of interest that were the result of a search.Use of other navigation map services such as TomTom, Navigon and Apple Maps.Support for a wider range of Public Means of Transport besides buses, including trains, subways and taxis.As it is common for the task performance to vary amongst users, or even for the same user, a method for adjusting to the user’s abilities is required to facilitate an efficient interface between the blind or visually impaired user and the navigation system. Currently, the interface is designed and fine-tuned exclusively for blind users, but we intend to experiment with high-contrast visual interfaces for partially sighted users as well.Providing the capability to repeat an issued instruction in the case where the user was unable to hear it due to external factors such as environmental noise or other distractions.Adding the capability to adjust the speed by which the overall brief description of the navigation route and the subsequent navigation instructions are issued.Allowing the user to flexibly change the destination without having to start the process all over again.Providing in-app updates. At the time of writing of this paper, the application is made available to download from an external link found in the bulletin board system of the Lighthouse for the Blind of Greece where manual installation is required.Multimodal sensory interface for traffic lights information by combining vibration and acoustic feedback. The above achieves safer and more accurate guidance for the user while passing traffic light crossings.

### 4.2. Limitations and Future Work for User Evaluation

We also acknowledge that the findings obtained from the Usability and UX evaluation of our application do not aid in generalizing as the participants are only from Greece and are few. Various factors contributed to the latter, with the main ones being the difficulty in finding and recruiting for the interviews many people with severe visual impairment due to the COVID-19 pandemic and other challenges related to mobility issues. Recruiting participants around the globe, which could be facilitated through World Wide Web communities, would certainly lead to more general and sound conclusions and is a future endeavor. In addition, as part of the above effort, the participant pool will expand and be diversified in key areas such as age, gender, ethnicity, living environment and previous experiences with the regular white cane and other assistive technologies. This will provide an adequate basis for developing a better appreciation of the needs and requirements of a wider part of the blind and visually impaired global population. Finally, this will create opportunities to enhance and improve the acceptance and usage of the proposed and other related technologies by creating an extended version of the TAM model. Nonetheless, all is not for naught as the requirements gathered during interviews can form a useful basis for researchers in the field and for developers of related applications even though they express personal preferences, opinions and suggestions from this local group. This can be based on the assertion that the core effects of vision loss on any blind person, regardless of their origin and region, are common, and therefore, it is reasonable to assume that their preferences, along with the solutions they choose to overcome them, overlap to some extent.

Another concern that emerged as a result of the evaluations was the issue of participant response bias, which is known to significantly affect the preferences about technological artefacts displayed in participants’ responses [82] and is often not taken into account when conducting field experiments with blind participants. Specifically, response bias is attributed to the perceived, by the participants, socio-economic characteristics of the person conducting the interview and the potential preferences of that person with respect to the subject of the study. In the interviews conducted as part of the project, the interviewer was a member of the research team which potentially may have contributed to increasing the bias in the responses. To avoid this situation, the part of the semi-structured interviews that requests feedback could be given in Braille so those blind users could have the opportunity to read the questions and assess the applications themselves during the evaluation phase. Furthermore, the UX questionnaire, which is accessible via Google Forms, will be handed out in Braille format as well so that participants with low digital sophistication will be included.

As part of our future work, we intend to fine-tune the existing version of the simulation app, which is developed as an interactive virtual navigation software solution that supports both Android smartphones and PCs. The end goal is to ease the engagement with this type of technology and increase the effectiveness of its usage since spatial information can be obtained indirectly (prior to navigation). The details of this simulation app will be described in depth in a future paper. Furthermore, we are considering designing, implementing and validating a VR application that hopefully will improve the trainability of the blind and visually impaired.

### 4.3. Future Work for Ameliorating the General Trend on Adoption Rates Usage

Last but not least, the requirements/expectations identified regarding the usability, functionality and trainability can be used for the creation of a framework that extends the Technology Acceptance Model to blind or visually impaired individuals. Such an extension will have to address the following two factors:(a)the increased sensitivity of the target group to needs related to their disability, and(b)the corresponding psychological patterns that stem from the insecurity caused by their disability.

The development of such a model is of paramount importance as it will make it possible to understand what creates positive anticipation. The latter plays a critical role in enhancing acceptance and continuing the usage of assistive technologies, otherwise blind users most often resist trying anything new until it is required.

This behavior inhibits assistive technology adoption, and, in a future paper, there will be a demonstration of how these limiting factors can be reduced by appropriate user-centered training models. The paper’s focus will be to provide a thorough description of the training models regarding the use of applications and how these produce guidelines that contribute to the creation of a revised version of TAM targeting the difficulties that affect the blind and visually impaired people. Furthermore, we will present, in the context of our newly proposed extension, a comparative evaluation of several various navigation methods that include both our application and other traditional methods, such as the use of a white cane with or without an accompanying guide dog. Finally, part of the presentation will be the evaluation of other existing applications currently favored by most blind users in terms of our proposed extended TAM model.

## 5. Conclusions

This paper has outlined the development and the extended Usability and UX evaluation of a specialized mobile outdoor pedestrian navigation system for people that are blind and visually impaired. The system was developed on the Android platform following a cognitive design process that incorporated extended interviews with blind users who had expertise in assistive technologies, as well as blind users with low digital sophistication, and resulted in a detailed requirements elicitation. Our functional prototype system, besides the Android device, comprises a high-accuracy GPS tracking sensor, an ultrasonic sensor for the detection of near-field obstacles along the route and a second external device mounted on traffic lights that monitors their status. The system interacts in real-time with the environment and issues navigational and object detection instructions, as well as traffic light crossing instructions. The content of the orientation instructions is customized to the special needs of our target group and are the same ones that are being taught in the O&M courses.

Overall, user experience was positively evaluated by all participants, however, they expressed an expectation for the system to improve its adaptability by providing more customizable options that suit better the users’ personal preferences. These include the available virtual assistants, more settings for the Talkback service and the voice interface with the Android device. Further enhancing its adaptability will, subsequently, improve the positively evaluated usefulness of our application. The users already report benefits in their lives as the application removes restrictions concerning pedestrian navigation. A consequence of the changes applied to the Adaptability features of our application will create the need for horizontal changes in the learning process. The system was found both dependable, due to the reliable and predictable nature of the app’s operations, and trustworthy, due to the quality of the information provided during navigation. The users also found the system to be efficient, as it enables them to swiftly complete their tasks, while they had, on one hand, mixed feelings about how explicit and straightforward it was to learn the application and, on the other, about the non-functional characteristics of the issued instructions that could potentially improve the perceived quality of service.

The data collected from the usability evaluation showed a high completion rate and a small number of cases where the participants could not successfully complete the given task. Nonetheless, throughout this process, we discovered some of the limitations of our current implementation that could potentially be improved. Although we have managed to deliver a functional version of the obstacle detection system [59], there are still scenarios, such as balconies or short trees, that need to be improved in the future. Furthermore, users quite often forgot to activate the public means of transport mode of operation that tracks the route on the bus. Another issue that became apparent was the fact that sometimes users would not react to the audio feedback related to traffic light crossings due to noise from the environment. Both of the last two cases will be addressed in the future by providing suitable haptic and audio feedback.

In response to the crucial requirement for trainability concerning the use of our proposed app, we developed a supplementary training version, identical in functionality to the main application, that familiarizes the user with the various features. The great need for trainability is due, on the one hand, to the fact that the skills, ease of learning and familiarity with digital platforms differ greatly between users and, on the other hand, the challenges that arise as a result of the interaction with a dynamic and unpredictable environment are more demanding for the blind people. The most effective way of solving the above difficulties is via the simulation of navigation routes utilizing familiar equipment that enables the users to experimentally navigate by replaying routes at their pace and place. Additionally, it is made to be convenient for the users as they are not required to carry all the standard equipment for regular navigation.

Intending to create an effective training tool, we identified the gaps that arise when people learn about the routes and their surroundings by reviewing the literature. In practice, various in situ navigation aids, tactile (and interactive) maps, as well as virtual navigation solutions that require special equipment, are used as means of training. These methods entail time-consuming and costly processes that our training version tool avoids. Furthermore, it allows for the combination of the positive aspects of the field and lab tests as it protects the users from the hazards of trials in real scenarios. Additionally, after the completion of the pilot tests, users were asked to evaluate their experience with the application in combination with the educational process. In general, most of the users evaluated the above process positively. Specifically, in a short time, they became familiar with the application environment.

Finally, our gained experience from the validation and evaluation of the proposed system set a starting point to surpass barriers that society imposes on people with disabilities, according to the Social Model of Disability [83].

## Figures and Tables

**Figure 1 sensors-22-04538-f001:**
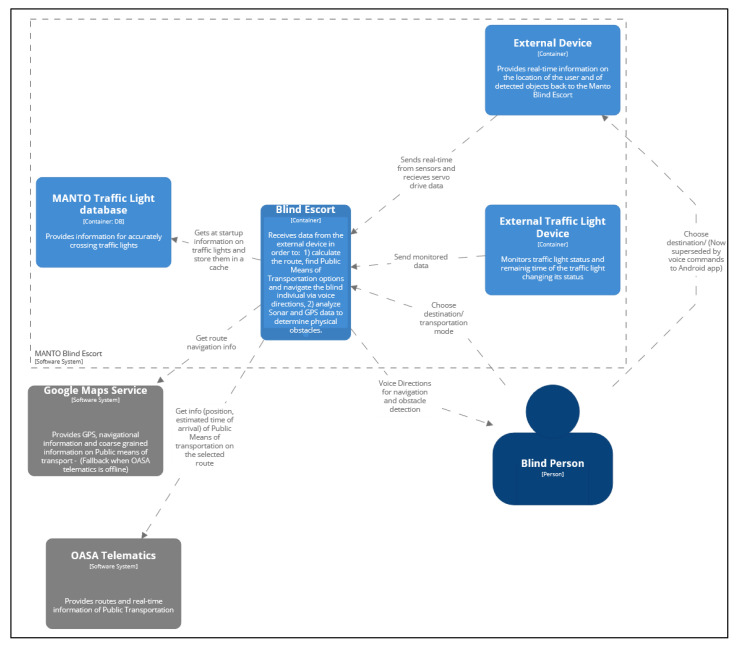
Architectural diagram of the application.

**Figure 2 sensors-22-04538-f002:**
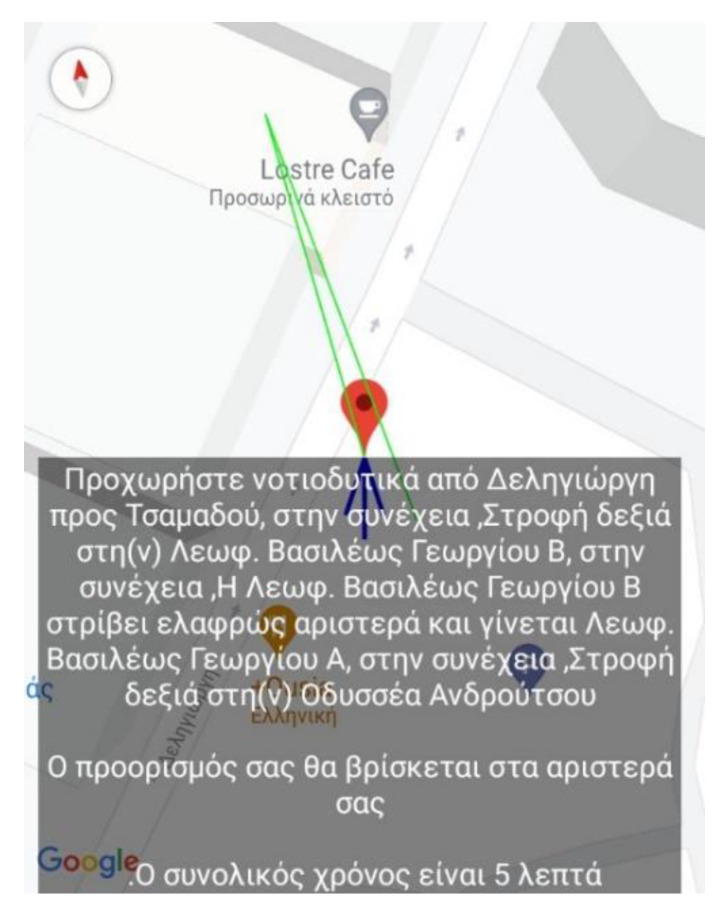
Navigation with great accuracy—text in figure: “Head southwest on Deligiorgi toward Tsamadou. Next, turn right onto Leoforos Vasileos Georgiou B. Next, slight left onto Leoforos Vasileos Georgiou A. Next, turn right onto Odissea Androutsou. Your destination will be on your left. Overall estimated time is 5 min”.

**Figure 3 sensors-22-04538-f003:**
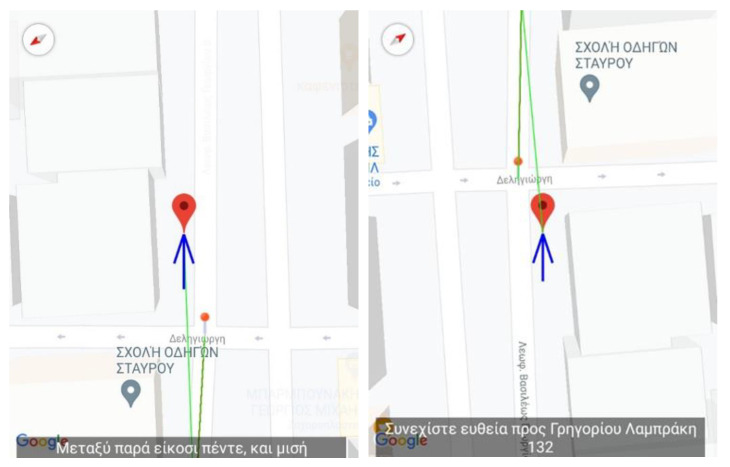
(**left**–**right**). Repositioning the user back to the correct navigational route—text in left figure: “Between 6 and 7 o’clock”; Text in right figure: “Continue onto Grigoriou Lampraki 132”.

**Figure 4 sensors-22-04538-f004:**
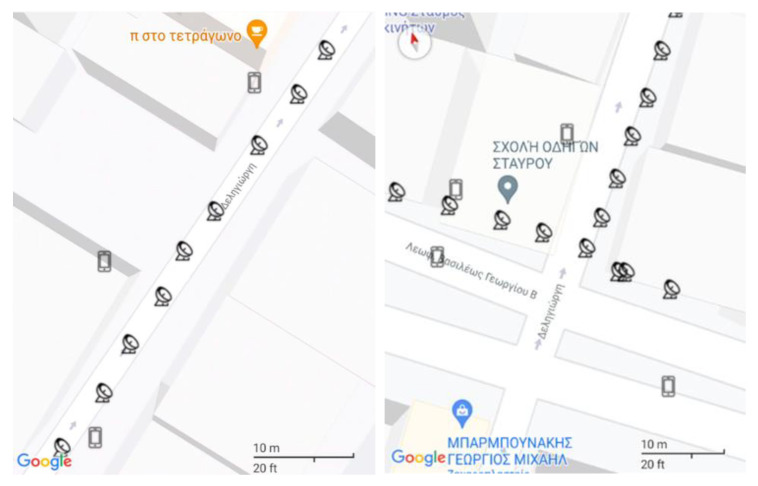
(**right** and **left**). Great accuracy and tracking density.

**Figure 5 sensors-22-04538-f005:**
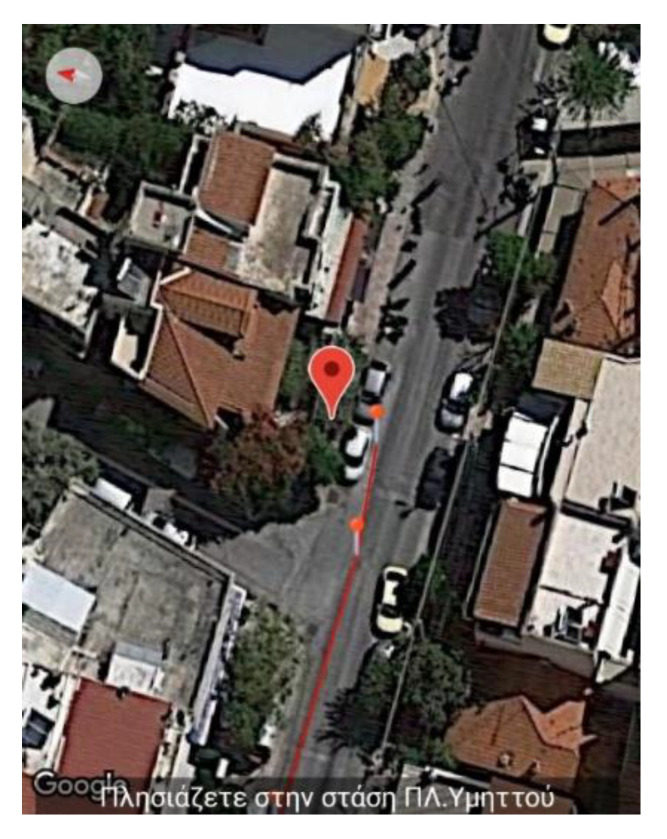
The application issues the instruction “Approaching Ymittos bus stop”.

**Figure 6 sensors-22-04538-f006:**
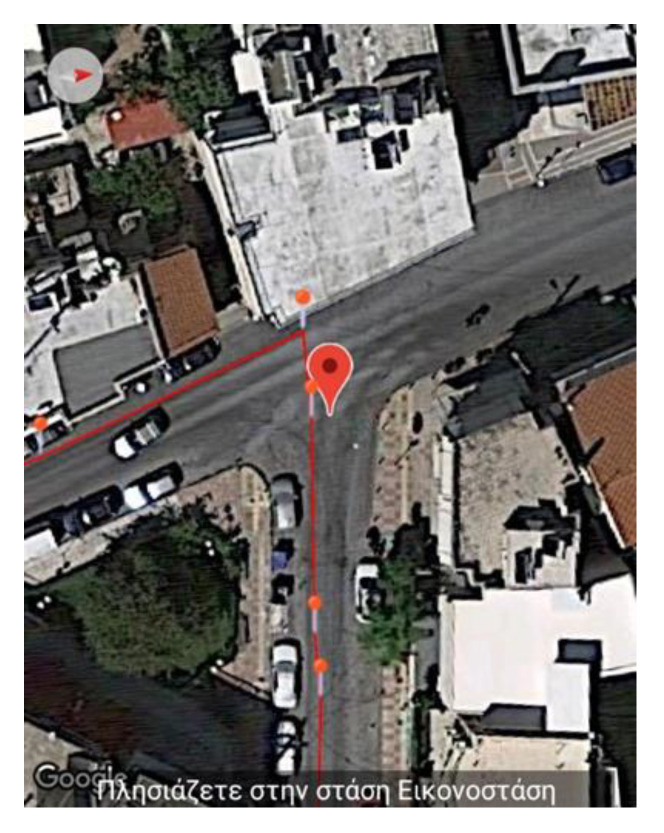
The application issues the instruction “Approaching Eikonostasi bus stop”.

**Figure 7 sensors-22-04538-f007:**
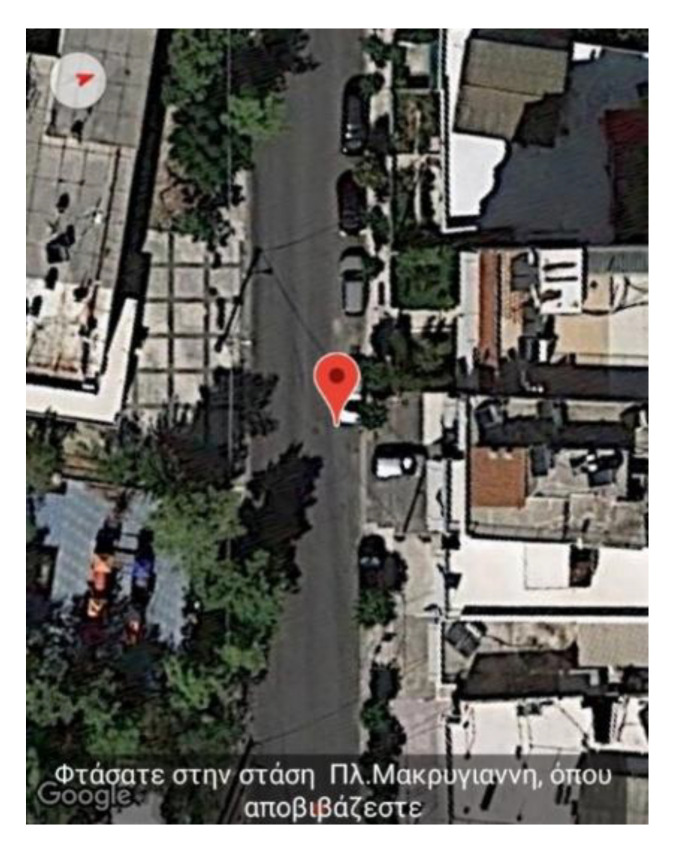
The application issues the instruction “You reached Makrygianni Square stop—Exit the bus”.

**Figure 8 sensors-22-04538-f008:**
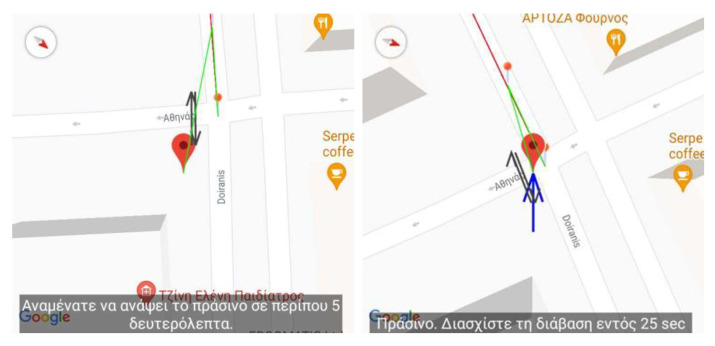
(**Right** and **left**). Passing traffic lights crossings with safety—text in left figure: “Five seconds remaining for the traffic light to turn green”; Text in right figure: “Traffic light turned green. 25 s remaining to cross”.

**Figure 9 sensors-22-04538-f009:**
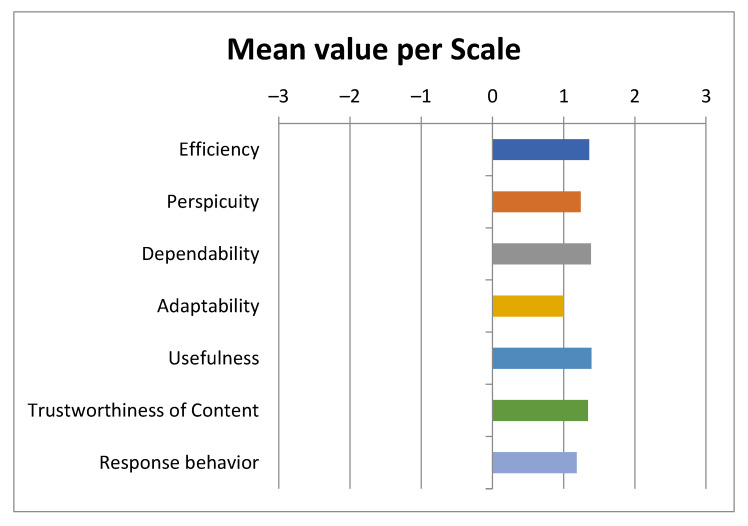
Mean value per Scale.

**Figure 10 sensors-22-04538-f010:**
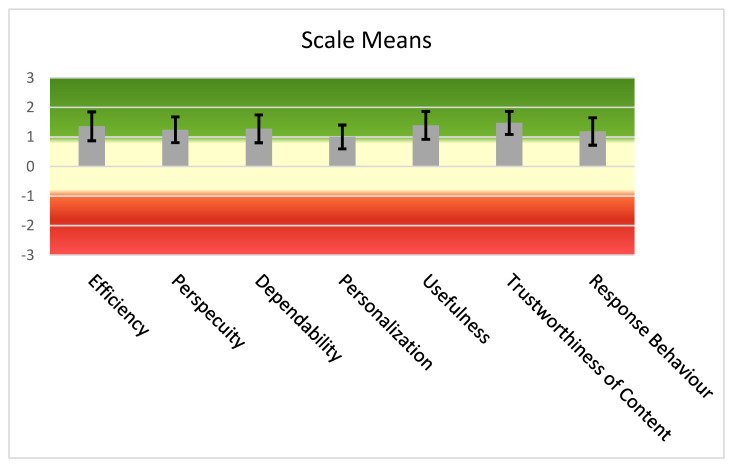
Scale Means and Standard Deviation.

**Figure 11 sensors-22-04538-f011:**
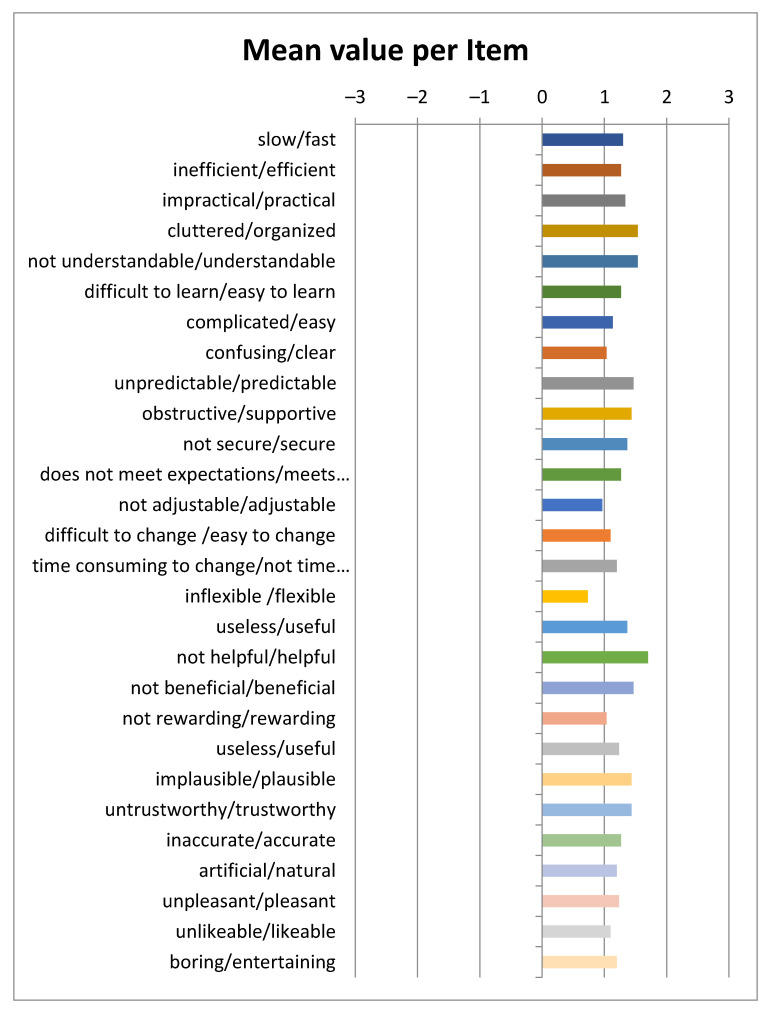
Mean value per Item.

**Figure 12 sensors-22-04538-f012:**
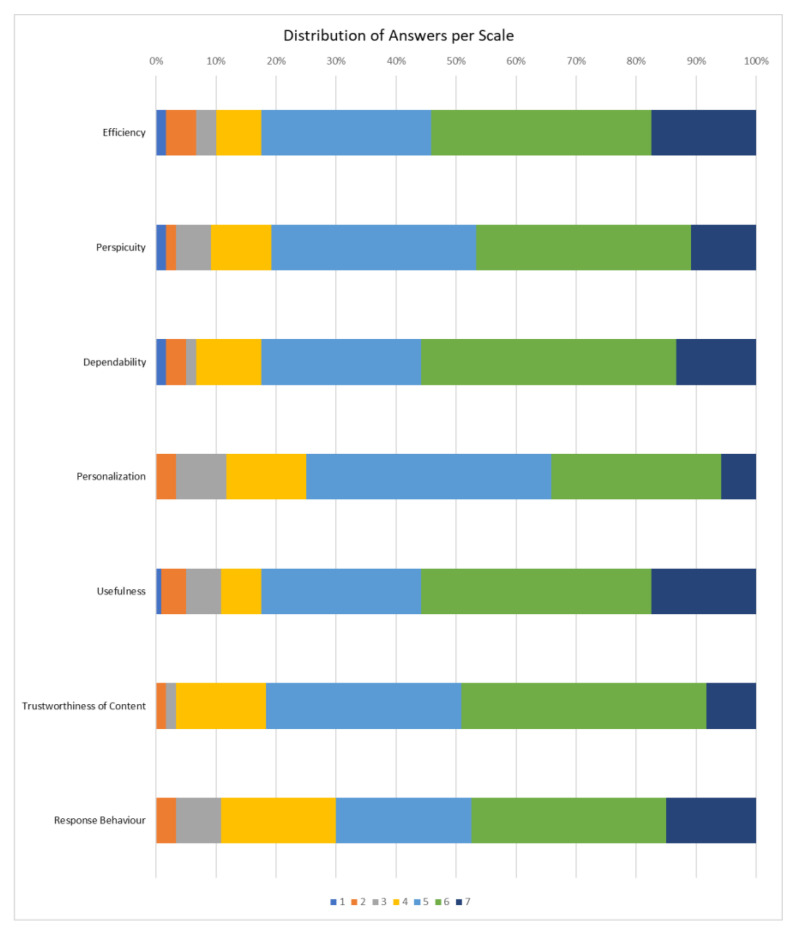
Standard Deviation of scales.

**Figure 13 sensors-22-04538-f013:**
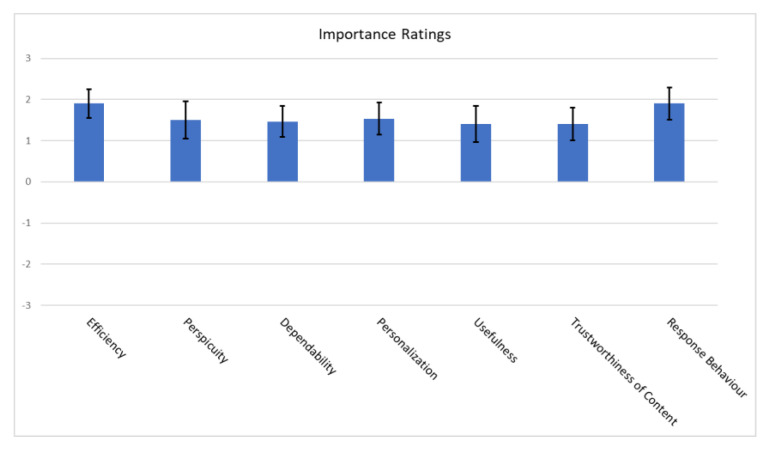
Importance ratings of scales.

**Figure 14 sensors-22-04538-f014:**
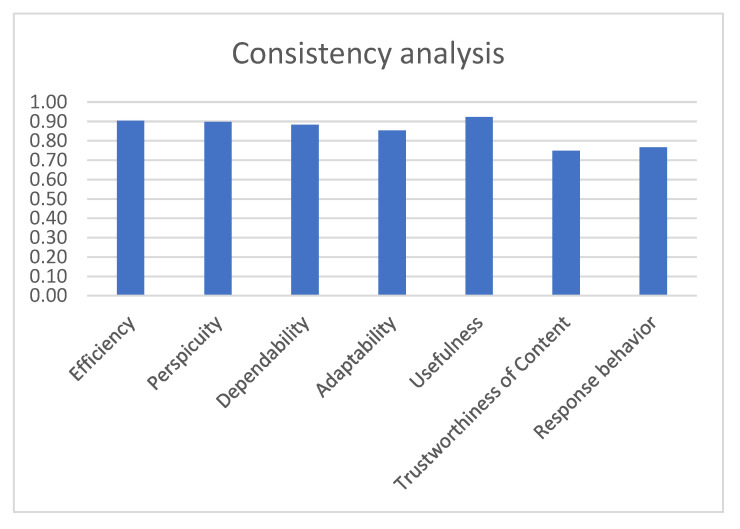
Consistency Analysis.

**Table 1 sensors-22-04538-t001:** Participants’ Characteristics.

	Gender	Age	Degree of Visual Impairment	Cause of Vision Loss	Digital Sophistication
P1	Male	55	Complete	By birth	High
P2	Female	35	Severe	By birth	Average
P3	Male	36	Complete	Diabetes	High
P4	Male	40	Almost complete (95%)	By birth	Low
P5	Male	40	Almost complete (95%)	By birth	Low
P6	Female	55	Complete	Retinopathy (28 years old)	Low
P7	Male	40	Almost complete (90–95%)	By birth	Low
P8	Male	40	Complete	Cancer (7 years old)	Low
P9	Male	35	Almost complete (>95%)	Benign tumor (15 years old)	Low
P10	Male	60	Complete	By birth	High
P11	Male	30	Complete	By birth	High
P12	Male	40	Complete	By birth	High
P13	Male	38	Almost complete (90–95%)	Craniocerebral injuries at 23	High
P14	Male	54	Complete	By birth	High
P15	Female	39	Severe	By birth	Average
P16	Male	36	Complete	Diabetes	High
P17	Male	46	Almost complete (95%)	By birth	High
P18	Male	44	Almost complete (95%)	By birth	Low
P19	Female	52	Complete	Retinopathy (23 years old)	Low
P20	Male	50	Almost complete (90–95%)	By birth	Low
P21	Male	40	Complete	Cancer (15 years old)	Low
P22	Male	35	Almost complete (>95%)	Benign tumor (6 years old)	Low
P23	Female	60	Complete	By birth	Low
P24	Male	47	Complete	By birth	Low
P25	Male	49	Complete	By birth	Low
P26	Female	38	Almost complete (90–95%)	By birth	Average
P27	Female	65	Complete	By birth	Average
P28	Female	39	Complete	By birth	Average
P29	Female	37	Complete	By birth	Average
P30	Female	40	Almost complete (90–95%)	Diabetes	Low

**Table 2 sensors-22-04538-t002:** Completion Rate.

Participant	Completion of a Route	Entering and Exiting the Bus	Passing Crossing with a Traffic Light
P1	3	1	0
P2	3	1	1
P3	3	1	0
P4	4	2	1
P5	2	2	0
P6	3	1	1
P7	3	2	0
P8	2	1	1
P9	3	2	1
P10	3	2	1
P11	3	2	0
P12	4	1	1
P13	3	2	0
P14	3	1	1
P15	4	1	1
P16	3	1	1
P17	3	1	0
P18	3	2	1
P19	3	2	0
P20	4	2	1
P21	3	1	1
P22	3	2	1
P23	4	2	0
P24	3	2	1
P25	3	1	1
P26	3	2	1
P27	4	1	1
P28	3	2	1
P29	3	1	1
P30	4	2	1

**Table 3 sensors-22-04538-t003:** Pedestrian Navigation Errors.

Participant	Collision with Obstacles	Veering	External Factor	Missed Turn	Over-Turn	Issued Instructions	Assisted
P1	3	1	1	1	1	1	1
P2	2	2	1	2	1	1	1
P3	1	0	1	1	1	1	1
P4	0	0	1	1	0	0	0
P5	2	0	1	1	0	0	2
P6	3	1	1	0	1	1	1
P7	2	1	0	0	1	1	1
P8	2	0	0	1	0	0	1
P9	1	3	2	1	0	1	1
P10	0	0	1	2	1	1	1
P11	3	0	1	1	1	1	1
P12	1	0	1	0	1	0	0
P13	1	1	1	1	0	1	1
P14	1	2	1	0	1	1	1
P15	1	2	1	1	2	0	0
P16	2	3	1	0	2	1	1
P17	3	1	0	1	2	1	1
P18	2	1	0	0	1	1	1
P19	2	2	1	1	1	1	1
P20	1	0	1	1	3	0	0
P21	1	0	1	1	1	1	1
P22	0	3	1	0	1	1	1
P23	0	0	0	1	1	0	0
P24	1	1	1	1	1	1	1
P25	2	1	1	1	1	1	1
P26	2	0	1	0	0	1	1
P27	2	0	0	0	0	0	0
P28	1	0	0	1	1	1	1
P29	1	1	1	1	1	1	1
P30	1	0	1	1	1	0	0

**Table 4 sensors-22-04538-t004:** Traffic lights crossings and public means of Transport Navigation Errors.

Public Means of Transport					Traffic Lights Crossings		
Participant	Boarding Button	Boarding	Exiting	Assisted	Veering	Reaction Time *	Assisted
P1	3	1	1	1	1	1	1
P2	2	0	0	1	1	1	0
P3	1	1	0	1	1	1	1
P4	0	0	1	0	0	0	0
P5	2	1	0	0	1	1	1
P6	3	0	1	1	0	0	0
P7	2	0	0	1	1	1	1
P8	2	0	1	1	1	0	0
P9	1	0	0	0	0	0	0
P10	0	0	0	0	0	0	0
P11	3	0	0	0	1	1	1
P12	1	1	1	1	1	1	0
P13	1	1	0	0	0	0	1
P14	1	1	1	1	1	1	0
P15	1	1	0	1	1	1	0
P16	2	1	1	1	0	1	0
P17	3	0	0	1	1	1	1
P18	2	0	0	0	0	0	0
P19	2	1	1	0	0	0	1
P20	1	0	1	0	0	0	0
P21	1	1	0	1	1	1	0
P22	0	0	0	0	0	0	0
P23	0	0	0	0	0	0	0
P24	1	1	0	1	0	1	1
P25	2	1	0	1	1	0	0
P26	2	1	1	1	0	1	0
P27	2	1	1	1	1	1	0
P28	1	0	0	0	0	0	0
P29	1	0	1	0	1	1	0
P30	1	0	0	0	0	0	0

Reaction time *—this refers to the reaction time to the status change notifications.

**Table 5 sensors-22-04538-t005:** Mean value, standard deviation and confidence intervals of the selected scales.

Scale	Mean Value	Standard Deviation	Confidence	Confidence Intervals	
Efficiency	1.36	1.37	0.49	0.87	1.85
Perspicuity	1.25	1.22	0.44	0.81	1.68
Dependability	1.38	1.25	0.45	0.94	1.83
Personalization	1.00	1.13	0.41	0.59	1.41
Usefulness	1.39	1.32	0.47	0.92	1.86
Trustworthiness of Content	1.34	1.00	0.36	0.99	1.70
Response Behavior	1.18	1.30	0.46	0.72	1.65

**Table 6 sensors-22-04538-t006:** Cronbach per scale.

Efficiency	0.9
Perspicuity	0.9
Dependability	0.9
Adaptability	0.85
Usefulness	0.92
Trustworthiness of Content	0.72
Response Behavior	0.77

## Data Availability

Not applicable.

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
