# Peer review of "An Extended Usability and UX Evaluation of a Mobile Application for the Navigation of Individuals with Blindness and Visual Impairments Outdoors—An Evaluation Framework Based on Training"

_sensors, 2022, doi:10.3390/s22124538_

Round 1

Reviewer 1 Report

This contribution is focused on a very interesting topic. It is well-written except for a few typos through the text.
Nevertheless, I have some comments that should be addressed before publication.

A comparison with the technologies currently employed as electronic travel aids for visually impaired people should be added, particularly in such a long article.

To this aim I suggest citing electromagnetic/radar technology, doi: 10.1109/JERM.2021.3117129 and camera technology doi: 10.3390/app12062802

A critical comparison between pros and cons of the different technologies used as aids for visually impaired people should be added.

The quality of Fig.1 should be improved and the font size increased.

Fig. 2, 3 are meaningless. This is also due to the non-English language.

What is the accuracy of the GPS sensor? It seems not adequate for such a kind of application.

Table 4 should be more compact.

Some discussion concerning the energy consumption of the different hardware solutions might be interesting for the readers.

An accurate comparison with the existing solutions within the scientific literature should be provided to highlight the novelty of the contribution.

Author Response

Thanks for your kind suggestions. Please see the attachment.

Reviewer 2 Report

The manuscript entitled “BlindRouteVision: An extended UX evaluation of an outdoor pedestrian navigation mobile application for the blind and visually impaired in the context of user-centered training models” involves an outdoor pedestrian navigation mobile application for blind and visually impaired people. However, it is poorly structured and does not look like an academic paper that emphasizes scientific issues and innovations. The manuscript needs to be completely revised before further assessment, and the following suggestions should be considered during the revision.

  1. Since the type of the manuscript is Article, please write the manuscript as a paper dealing with scientific or technical problems, not as the present "Project Concluding Report".
  2. Please try to follow the format of the Sensor submission guidelines, in fact, both Section 1 and Section 2 of the manuscript are part of the Introduction. Please download the Sensors-template (Microsoft Word template or LaTeX template) at the “Instructions for Authors” of Sensors, which details the sections that can be used in a manuscript.
  3. The title of an article should be a highly condensed summary of the research content, please rewrite the title, and since “BlindRouteVision” is a word or phrase that is invented in this manuscript, it is not suitable for the main title.
  4. The authors are strongly encouraged to use the style of structured abstracts that is suggested in the Sensors-template.
  5. Please list five or so pertinent keywords specific to your article yet reasonably common within the subject discipline.
  6. The Conclusion is unusually long, please condense it.
  7. Please highlight creative works or works that were done in this study during the revision.
  8. Please reduce the number of references cited from the current 114 to around 50.

Author Response

(The authors gave the same response as above.)

Round 2

Reviewer 1 Report

The authors addressed all my concerns. I have only a small comments concerning the added references. You wrote "We extended the number of papers presented in the background section" and "This paper is now part of the literature review". However the added reference seems an old version of the suggested 10.1109/JERM.2021.3117129. Therefore I suggest to add also this reference to update the list with most recent references.

Author Response

Dear Reviewer,

We followed your suggestion and we included the proposed version of the paper. 

Reviewer 2 Report

The manuscript has been moderately revised, but the following two suggestions still need to be further considered.

1.     The reviewer still felt that the paper is lengthy, more or less, thus it is suggested that the paper should be further condensed.

2.     The references the authors cited in the paper must be further selected. Among 122 papers, there are at least about 50 papers that are not of interest to potential readers should be deleted.

Author Response

Dear reviewer, 

We tried to address your concern but it is impossible to condense the paper in the provided time period. We consider that the paper as is best reflects our work. However, we acknowledge your concerns. We could either remove a few references (approximately 15), which we think cannot condense enough the size of the paper or we could remove as a last resort the trainability section and the relevant simulation tool (in our opinion this enhances our UX evaluation), which saves approximately 5 pages. We hope that this is a compromising solution without requiring extensive editing that is impossible to perform in two days.  

Best Regards, 

The authors